# Sustainable Urban Mobility Planning in the Port Areas: A Case Study

**Marinko Maslaric** [1] , **Sanja Bojic** [1] , **Dejan Mircetic** [2], **Svetlana Nikolicic** [1] **and Ranka Medenica Todorovic** [1,*]

1  Faculty of Technical Sciences, University of Novi Sad, 21000 Novi Sad, Serbia; marinko@uns.ac.rs (M.M.); s_bojic@uns.ac.rs (S.B.); cecan@uns.ac.rs (S.N.)
2  Institute for Artificial Intelligence Research and Development of Serbia, 21000 Novi Sad, Serbia; dejan.mircetic@ivi.ac.rs
*  Correspondence: ranka.medenica@uns.ac.rs

**Abstract:** Sustainable development, urban planning, mobility, and transport planning, integrated within the context of sustainable urban mobility, have been central themes in both scientific and applied spheres over the past few decades. In port cities, it becomes particularly essential to tackle sustainability issues given the pollution and noise emanating from ships and other port-related activities. To meet mobility and transportation sustainability needs in the port area, a port should implement measures aligned with a sustainable urban mobility planning (SUMP) approach. However, many ports have thus far achieved limited results in this direction due to the absence of an approach to defining sustainable mobility solutions based on the SUMP approach for an urban area associated with the given port. The overall aim of this paper is to support the development of territorial SUMP for port areas by proposing a methodology that identifies and prioritizes sustainable mobility solutions tailored to a specific port area. The proposed methodology is applied in the Port of Bar (Montenegro) through an appropriate case study. In this case study, the methodological steps are systematically followed, resulting in the practical implementation of the selected mobility solution: the use of a hybrid bus for internal employee transportation within the port area. The undertaken case study underscores the simplicity, practical applicability, and adaptability of the proposed methodology.

**Keywords:** ports; port area; SUMP; micro-SUMP; case study; hybrid bus

## 1. Introduction

Sustainability is a topic of the utmost importance, discussed at the state, institutional, and enterprise levels. Key sustainability concerns for cities encompass sustainable transport, architecture, spatial development, energy usage, and more [1]. It can be argued that mobility and transport, in particular, play a crucial role in achieving sustainable cities and communities [2]. Efficient mobility systems can reduce congestion, accidents, noise, pollution, and greenhouse gas emissions. According to the European Commission (EC) and the *Green Paper on Urban Mobility*, urban transport in the European Union (EU) is responsible for nearly 40% of the $CO_2$ emissions in the overall transportation sector and 70% of emissions of other pollutants [3]. The same paper reports that 69% of road accidents occur in cities, and approximately 1% of EU GDP is lost due to urban congestion. These statistics may be attributed to the fact that urban areas are continuously attracting more people. In 2018, 74% of all Europeans lived in towns and cities, and this proportion is expected to increase in the coming decades [4]. The growing population, urban sprawl, and longer commuting distances that extend beyond municipal boundaries exacerbate the demand for urban mobility, further intensifying congestion, environmental concerns, and social disparities. In an effort to address these challenges, the EC introduced the concept of sustainable urban mobility planning (SUMP) with its 2013 Urban Mobility Package [5].

In port cities, it is especially crucial to address sustainability concerns due to the pollution and noise generated by ships and other port activities. Notably, two-thirds of

the world's major ports are situated in clearly urban areas [1]. Ports play a vital role in the development of cities, while cities, in turn, provide essential resources and infrastructure for ports. This symbiotic relationship means that ports utilize urban spaces, infrastructure, labor markets, and other resources. Consequently, it is imperative for ports to engage in close consultation and cooperation with city governments and other stakeholders to work towards ensuring economic prosperity as well as the long-term enhancement of the quality of life in the port area and the urban community it serves [6]. To address the existing sustainability needs in terms of mobility and transportation, both within the entire city area and the integration of the port area itself and its hinterland, a sustainable port should propose and implement various measures, actions, and strategies. These should align with a SUMP approach to ensure environmental protection, climate change mitigation, and the adoption of environmentally friendly technologies and renewable energy sources. In the context of ports as vital transport, logistics, and economic hubs, the creation of a SUMP thus takes on heightened importance. The number of studies and research papers providing insight into the development and implementation of SUMP in port cities and addressing the unique challenges and opportunities that arise in these urban areas with significant port activities is limited. For example, paper [7] examines SUMPs and the pilot or permanent implementation of mobility measures in six Mediterranean port cities, whereas paper [8] delves into the topic of integrating sustainability efforts between the port and the city through the examination of a port area master and city mobility plan.

This paper outlines an approach and methodology underpinning the development of a territorial SUMP tailored to the unique needs and circumstances of a port and its associated urban areas. The contemporary perspective of ports views them as industrial areas playing a significant role in providing value-added services. This perspective guides their transformation into special economic areas that demand enhanced accessibility for the workforce. In addition to the movement of commuters influenced by the interactions between the port and its surrounding territory, the additional types of mobility within the port area include the mobility of passengers during the initial and final legs of coastal shipping journeys by ferry as well as the mobility of tourists originating from cruise ships. Achieving a high level of availability of the port area for these types of mobility can be realized through an integrated and sustainable transport system. This is a highly complex task because, despite their interconnectedness, ports and cities are typically governed by distinct bodies. This disparity makes joint mobility planning challenging, and this is the issue that this paper tries to address. By proposing a general approach and methodology that facilitates the collaboration between port authorities and municipalities, it contributes to creating the foundation needed for integrated and sustainable mobility planning. Furthermore, the developed methodology can support port authorities and, more broadly, decision-makers from both public and private sectors engaged in the overall process of port sustainability development.

The proposed methodology is based on several steps involved in crafting a comprehensive territorial SUMP, with a particular emphasis on an integrated view of the port area with its hinterland, respecting both the sustainable development of the urban city area and the port area. The proposed approach involves a two-stage mobility planning strategy: the first stage encompasses a city-wide SUMP, with its definition falling under the responsibility of the municipality, and the second entails a micro-SUMP designed specifically for the port area and formulated as a result of the collaboration between the municipality and port authorities. Therefore, this micro-SUMP is seamlessly integrated with the broader city-wide SUMP. The primary contribution of the developed approach is founded on the principle of subsidiarity, which involves the creation of a fundamental SUMP, within which the formation of a micro-SUMP dedicated to specific areas is encouraged. These micro-SUMPs enable local-level actions tailored to address area-specific issues. In the case of a port area, this approach ensures that mobility and sustainability planning can be customized to the unique challenges and opportunities that pertain to that specific region. The utility of the

developed approach was assessed through an appropriate case study for the formulation of sustainable solutions related to mobility in a certain port area.

## 2. Literature Review

### 2.1. Sustainable Urban Mobility Plan (SUMP)

Drawing from established planning practices and principles, the concept of SUMP introduces an innovative perspective on urban mobility planning. Unlike conventional transportation planning, which primarily focuses on organizing movement and providing infrastructure for particular transport modes, SUMPs place a strong emphasis on facilitating the movement of people within an urban environment. Introduced as a formal framework for strategic transport planning, officially proposed by the EU and endorsed by the EC in relevant documents, the SUMP serves as an instrument aimed at improving urban accessibility and ensuring dependable, high-quality mobility and transportation. Simultaneously, it contributes to the achievement of the EU's climate and energy targets. Over the past 15 years, the development of SUMPs in Europe has been driven by the EC's initiatives, particularly the *Urban Mobility Action Plan*, published in 2009 [9], the 2011 *White Book on Transport* [10], and the 2013 Urban Mobility Package. These plans, largely promoted by the EC, serve as strategic, long-term, and sustainable visions for urban areas, primarily funded through European structural and investment funds. The SUMP approach is versatile and applicable across a wide range of urban area sizes, from towns to metropolitan regions [11].

Numerous guidelines, often at the supranational or national level, provide a structured framework for developing effective SUMPs. These guidelines typically involve stages such as status analysis, vision development, objective and target setting, policy and measure selection, active communication, monitoring and evaluation, and the identification of lessons learned [12]. One of the earliest and most widely used sets of guidelines [13] featured 11 steps organized into 31 sequential actions, emphasizing a cyclical process with concurrent activities. In 2019, a second edition of the SUMP guidelines was released [14], introducing the twelve steps of the "SUMP cycle" (Figure 1), which consists of four main phases and a total of 32 specific activities [14], allowing for concurrent operations and feedback loops.

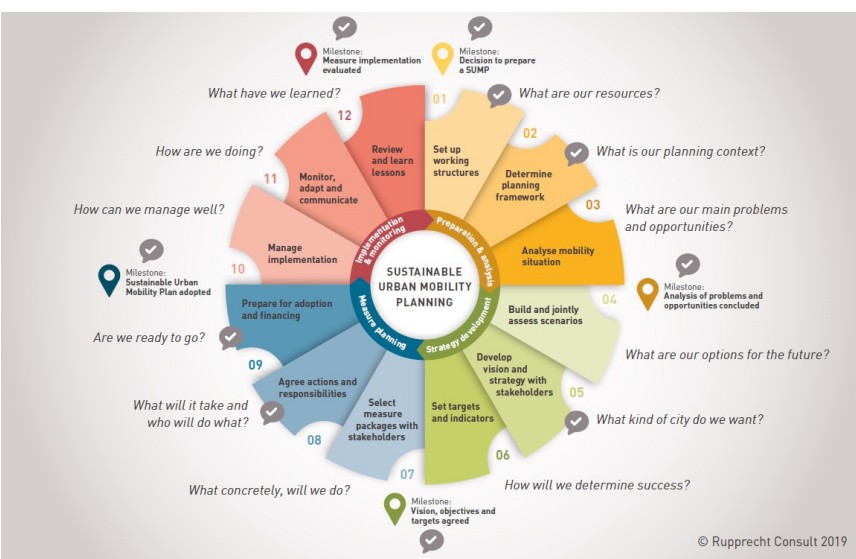

**Figure 1.** The twelve steps of SUMP—a decision maker's overview [14].

The central tenet of the SUMP concept is its unwavering focus on people. Regardless of their role, whether as commuters, business professionals, consumers, or any other capacity, the development of a SUMP revolves around "planning for people" [15]. This people-centric approach sets it apart from traditional transport planning, which tends to prioritize

traffic and infrastructure over the needs of individuals and their mobility. Benefits that can be achieved through a SUMP are numerous. Some of these benefits are tangible and quantifiable, while others are less so and may be challenging to assign a specific financial value. For instance, they may include social inclusion and a sense of civic pride associated with living in a green, successful, and sustainable environment.

A SUMP provides a strategic framework to integrate the concepts of smart and sustainable mobility, which are crucial for fostering efficient, environmentally conscious transportation systems. The city becomes smarter and more sustainable when it is competitive and preserves natural resources, encourages community participation, enhances the quality of life, integrates transport and information communication technologies (ITC), and promotes social and human capital [16]. Smart mobility extensively incorporates ICT in both backward and forward applications to support the optimization of traffic flows. This integration also serves to gather feedback from citizens regarding the livability of cities or the quality of their public transport services [17]. ICT is undeniably the keyword in defining mobility as "smart". As articulated in [18], smart urban mobility can be succinctly described as leveraging ICT to generate and share data, information, and knowledge that influences decisions, as well as utilizing ICT to enhance vehicles, infrastructures, and services and deriving improvements for all included stakeholders. The same authors stated that smart and sustainable need to be brought together to ensure their alignment, aiming to bridge technological (smart) and social (sustainable) considerations in optimizing the urban mobility system. Therefore, urban mobility policies tend to address aspects of sustainable and smart urban mobility by implementing appropriate strategic frameworks that should arise from a holistic and integrated approach to both urban and urban mobility planning. With the evolution of urban mobility policies, various cities, governments, and organizations have implemented diverse initiatives to shape that strategic framework for the future of modern, sustainable urban mobility [19]. These efforts culminate in SUMP, which not only optimizes urban mobility but also enhances accessibility, reduces environmental impact, and embraces the evolving landscape of smart and sustainable solutions.

As previously stated, the EC plays a leading role in developing guidelines for SUMPs and promotes the exchange of experiences among cities through various programs. On the other hand, national governments establish the overarching context within which cities formulate their SUMPs [20]. This context includes legislation that governs urban transport development and the regulatory framework that guides its operation. The literature contains a wealth of works on the subjects of SUMP development in European cities. For example, paper [21] discusses the development and implementation of a SUMP in Portugal in order to understand how climate change goals and equity issues in accessibility have been addressed through the first generation of SUMPs. It also outlines further research needs about the SUMP approach towards climate change. Different ways to promote the concept of SUMP among European cities are explored in [4], where the main findings suggest that EC takes different roles and uses all mechanisms in parallel, albeit with different intensities. A systematic overview of how various social groups interact in German SUMPs is provided in [22], together with valuable input for making methodological decisions when designing a participation strategy for a SUMP. Similar to the previous reference, in [23], the authors analyzed the European guidelines for the development and implementation of SUMPs in the context of Italian cities. The authors in [24] utilized Eurobarometer survey data that captures the perspectives on urban mobility to explore the primary factors influencing how frequently citizens use public transportation, particularly in three European countries: Germany, Italy, and the Netherlands. Their results suggest that a comprehensive set of socio-demographic, economic, and environmental aspects played a key role in determining urban travel behavior. Paper [25] focuses on examining the outcomes of 38 SUMPs conducted in Spanish cities. By comparing the results, the paper provides an overview of the evolution of efforts to create more sustainable mobility plans in Spain, along with guidelines for further improving the SUMPs.

A group of papers deals with the relationship between SUMP and the latest crisis with the idea that the COVID-19 pandemic and recent energy crisis represented an opportunity for governments to rethink their mobility systems and to develop or update SUMPs, taking into account these considerations [26,27]. Unlike the previously mentioned references, which primarily concentrate on the development of SUMPs in developed countries, there are also a smaller number of papers that address the development of SUMPs in developing countries [28–30]. This section should be concluded with the statement from [31] that while SUMP represents a general framework for defining sustainable mobility development in cities, applying measures from one city to another is challenging due to specific contexts and geographical and transport conditions. The same authors note that comparing SUMPs from different cities and countries is generally difficult, primarily because of the limited availability of SUMPs in non-native languages and variations in policy and planning levels for mobility.

*2.2. SUMP in the Port Cities*

A port city serves as not only a vital economic center but also a pivotal transportation hub. It seamlessly blends port-related industries, logistics, and industrial activities with the creative energies of its residents. Typically, a port city acts as a nexus for import and export trade, as well as a bridge between industry and tourism [32]. However, port cities face unique challenges when it comes to sustainable development, primarily due to their coastal locations, which expose them to the severe consequences of climate change. Sustainable development in port cities is demonstrated by the symbiotic relationship between the port and the city. The port contributes to the city's well-being by supporting trade, employment, economic growth, and environmental sustainability. Meanwhile, the ports depend on the city for its workforce [33]. Sustainable development in port cities necessitates a thoughtful and planned approach where, according to [34], transitional management can serve as a suitable method to analyze and plan the evolution of the sustainable relationship between the port and the city. According to the same authors, guiding port cities toward the desired direction of sustainability has become a key challenge that needs to be accepted and institutionalized at the local, regional, and national levels. Defining these guidelines for achieving sustainability in a port city is very challenging due to numerous contradictions related to situations where negative externalities resulting from increased industrial/trade/tourist activities cause environmental damages and economic, social, and cultural costs. On the flip side, port areas can become the entry point for the sustainable development of the whole urban system [35]. They present promising opportunities to experiment with innovative solutions addressing various sustainability issues, including energy transition, the development of innovative industries, cultural enrichment, and improved mobility [36]. According to the same reference, 'sustainable mobility' is among the ten goals that present the main challenges for sustainable port cities. This goal is aimed at enhancing mobility in the city port and alleviating congestion in both urban and port areas. This involves encouraging the development of soft, multimodal, and collaborative mobility and transportation options, implementing proximity-based urban logistics solutions, promoting the use of waterways, rail, and other non-fossil-based transportation modes in the city port area, and mitigating the adverse effects of peak activity periods in the city port through various means.

A comprehensive review of port sustainability is provided in [32], where the main categories of research focus are identified based on over 60 papers reviewed. It offers clear information on the geographic coverage and the main research topics, as well as directions for future research on port city sustainability. Sustainability in ports is often linked to the application of cleaner energy technologies [37] and the implementation of innovative digital technologies [38]. Regarding the issue of sustainable transport and mobility in ports, the majority of papers discuss this topic with a focus on freight transport associated with the economic and industrial activities of ports. For example, paper [39] discusses how the improvements in the traffic management policy and infrastructure of port cities can

help reduce port emissions and traffic congestion, while paper [40] explores how smart and sustainable logistics initiatives in port cities foster their transformation into nodes with efficient management of the flow of resources, emissions, materials, waste, people, information, and so on, that circulate within them.

Port cities face urban mobility challenges similar to those of other cities, but they also contend with increased traffic, transport, logistics, and tourism demands, often serving as regional or global hubs for both passengers and freight. Therefore, devising mobility plans for port cities, marked by notable traffic density and congestion issues, proves intricate due to the myriad and conflicting factors and needs involved. The situation is further complicated by political and financial challenges. The current modal split, combined with the port city's inability to manage the rapid expansion of port development and population growth, has the potential to worsen environmental and health issues for residents. Finally, establishing a sustainable urban mobility plan presents a significant challenge in terms of stakeholder participation, given the weak connections between port authorities and city authorities. Without cooperative and integrated initiatives between ports and cities, achieving sustainable mobility in port cities remains unattainable. Consequently, the need to develop sustainable mobility plans that provide an affordable, accessible, sustainable, and safe transportation system for all users is of paramount importance for these cities.

The literature contains numerous projects related to developing sustainable transport and mobility solutions for port cities. For example, within the framework of the EU Interreg Mediterranean Program, specifically the sub-theme of sustainable transport, seven modular projects (CAMPsUmp, EnerNETMob, LOCATIONS, MOBILITAS, MOTIVATE, REMEDIO, and SUMPORT), along with one horizontal project (GO SUMP), were approved and implemented between 2014 and 2020. These projects aimed to promote the widespread adoption of the SUMP concept throughout the Mediterranean regions, primarily through the development of SUMPs and the implementation of sustainable transport pilot services that have reached over 30 Mediterranean cities, encompassing initiatives such as electro-mobility networks, smart mobility systems, congestion mitigation strategies for tourist traffic, and the renovation of transportation infrastructure [12]. Despite their unique characteristics, varying sizes, populations, economic levels, and development stages, all these coastal cities with major ports share common challenges related to sustainable mobility planning. While some cities are well ahead and possess the economic resources and proactive measures to promote sustainable mobility (already having SUMPs in place or under evaluation), others significantly lag behind (lacking SUMPs altogether) [6].

Similar to the Mediterranean region, the EU co-funded project HUPMOBILE (Holistic Urban and Peri-urban Mobility) unites stakeholders to develop a comprehensive approach for planning and implementing sustainable mobility solutions in the Baltic Sea port cities. Partner cities, such as Turku (Finland), Riga (Latvia), Tallinn (Estonia), and Hamburg (Germany), are involved in planning, testing, and implementing innovative sustainable urban mobility solutions for both people and goods, with the aim of easy adaptability for other cities [8]. HUPMOBILE outputs include an overall framework presented as a modular online planning tool and various mobility concepts for external adoption. These plans and policies primarily focus on facilitating the shift from car-dependent mobility to more sustainable alternatives.

Another set of five European port cities, which had prior experience in the development, demonstration, and evaluation of innovative sustainable mobility solutions through the CIVITAS PORTIS project [41], includes Aberdeen (UK), Antwerp (Belgium), Constanta (Romania), Klaipeda (Lithuania), and Trieste (Italy). This project exemplified how sustainable mobility can enhance both functional and social integration between city centers and ports, promoting economic growth and enhancing the appeal of urban environments. This was achieved through the implementation of a total of 49 measures spanning four main categories: governance, community engagement, transportation systems, and freight management. These measures were aimed at encouraging people to adopt a more car-independent lifestyle by promoting active transportation (walking, cycling), public transit,

electric vehicles, and the utilization of intelligent transport systems and IT solutions to optimize multimodal transport, manage transportation routes, and introduce innovative modes of transportation.

## 3. Materials and Methods

### 3.1. Proposed General Approach to Micro-SUMP Development

As evident from the preceding literature review, the port and the city are not two separate entities; instead, they are closely intertwined, mutually interdependent, and mutually influential systems. They can be considered as nested systems. Port development contributes to the growth of cities, with many coastal cities having their origins in ports that supported their development. Conversely, the progress of the city's economy, technology, and transportation, in turn, promotes the development of ports [42]. As a result, the sustainability of port cities should be analyzed by considering the interaction and interdependence between these two systems—the port and the city.

This paper presents an approach and methodology that form the foundation for creating a territorial SUMP customized to the distinct requirements and situations of a port and the urban regions connected to it. The objective of such a territorial or micro-SUMP for the port area is to establish sustainable transportation guidelines that facilitate a harmonious connection between the port area and its surrounding region. This integration should take into consideration the principles of sustainable development for the city (including aspects such as spatial planning, traffic management, economic development, and social programs) as well as the port area. In line with this vision, the SUMP serves as a tool for aligning urban policies with the "sustainable city" concept and port policies with the "sustainable port" concept (Figure 2). Furthermore, the development of the micro-SUMP must also be in line with regional coordination and collaboration, as the transportation system under the purview of the micro-SUMP is intricately connected to the broader urban system and extends into regional and cross-border areas.

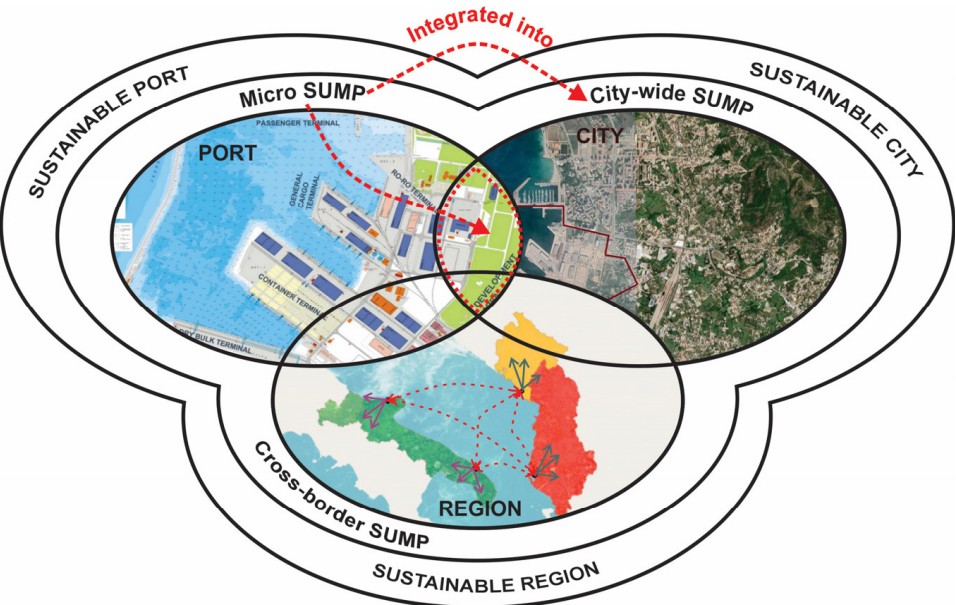

**Figure 2.** The relationships between different SUMP levels (created by the authors).

Therefore, this paper proposes a two-stage approach for developing and implementing a SUMP for the port area. The first stage involves a city-wide SUMP, while the second stage entails a micro-SUMP within the port area, which is integrated with the city's SUMP. This approach is rooted in the principle of subsidiarity within the SUMP, utilizing micro-SUMPs that address particular areas to effectively address localized issues. In this instance, the focus is on the unique challenges of the port area.

Justification for developing a micro-SUMP can be found in its ability to expedite the achievement of city-wide SUMP goals through a bottom-up approach. By focusing on specific city areas, decision-making is streamlined, thus reducing the time required for implementation. This allows residents, commuters, tourists, and workers in the targeted area to experience the benefits of the implemented measures sooner compared to broader city-wide measures. Furthermore, subsequent measures can be informed by the experience and adoption of earlier proposals. Analyses have also demonstrated the potential to promote the adoption of city-wide SUMPs by raising awareness of specific urban mobility issues in micro-locations. Providing location-specific evidence of the positive impacts of sustainable mobility measures can be persuasive in demonstrating the overall benefits of a SUMP for the entire municipality.

### 3.2. Methodology for Creating a Micro-SUMP for the Port Area

Respecting the adopted approach, the appropriate methodology for preparing the micro-SUMP is also proposed (as illustrated in Figure 3). This methodology is rooted in the SUMP guidelines outlined in [14] and primarily focuses on the preparation and establishment of SUMP's primary goals and objectives. The full implementation processes of the SUMP are beyond the scope of this paper. The methodology consists of two main parts: an analytical section and a proposal section. The analytical portion of the methodology encompasses the following steps, which aim to:

- Evaluate the SUMP concept in general and assess the current status and state-of-the-art of SUMPs, including their drivers and barriers;
- Assess the applicability of the SUMP to the city port by reviewing relevant guides, case studies, and projects;
- Describe the current state of mobility and the transportation system within the port area;
- Review best practices concerning practical, sustainable mobility solutions within the city port area.

The final steps in this analytical phase involve defining the primary SUMP objectives and laying the groundwork for potential strategies and practical solutions. These details are further elaborated in the proposal section of the methodology, which includes:

- A clear vision and strategic objectives;
- The establishment of specific objectives, practical measures, and necessary tools;
- An organizational framework for the implementation of the proposed pilot action.

The primary objective of this methodology is to facilitate the design and implementation of measures that align closely with the specific needs of the city area, particularly the port area. It is crucial to emphasize that the development and implementation of this methodology should not be viewed as a separate layer in the city's transport planning but should instead be integrated with its existing urban mobility plans and processes. This methodology should be an integral part of the city-wide urban mobility planning strategy since solutions that combine the sustainable development of both ports and city areas are essential.

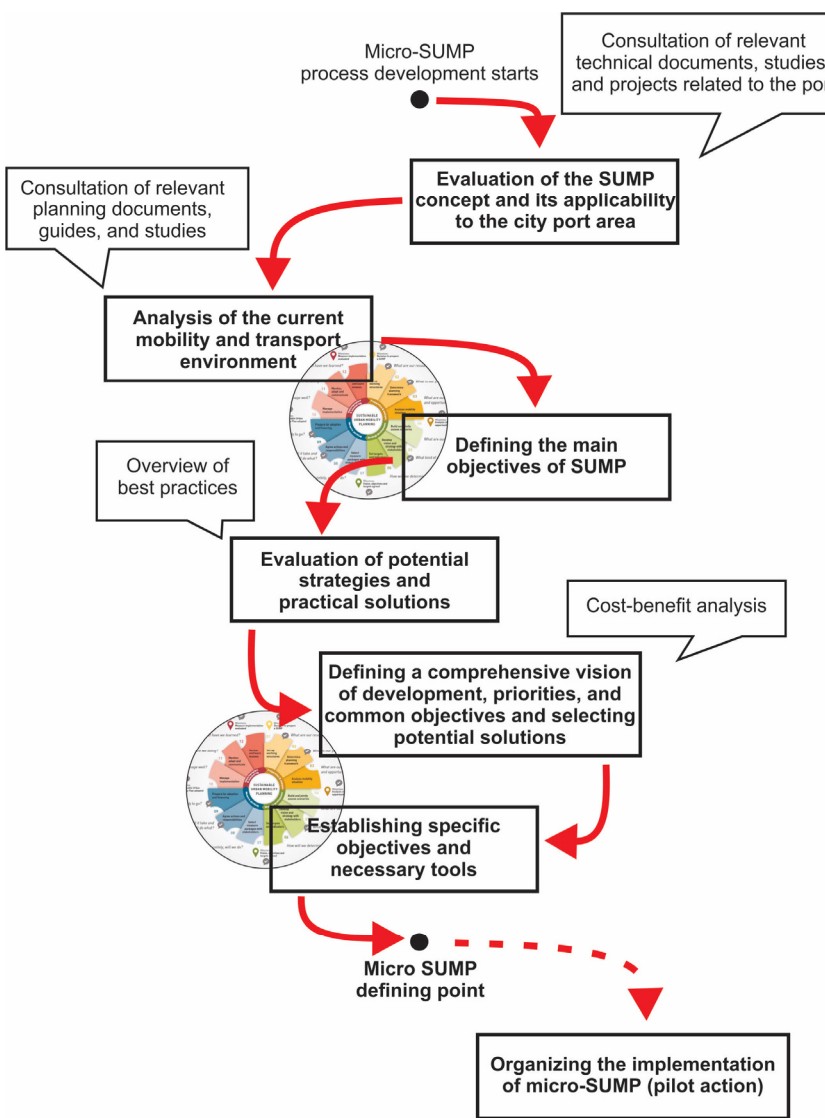

**Figure 3.** Methodological steps for micro-SUMP development (created by the authors).

## 4. Case Study

The effectiveness of the proposed methodology for creating micro-SUMPs in port areas was assessed using a qualitative single case study. A case study as a research method is suitable to understand how and why a phenomenon takes place in specific situations [43]. We employed a single case study to perform a thorough examination of the methodology for developing a micro-SUMP for the Port of Bar in Montenegro. The credibility of this case study relies on the transparency of the proposed approach and methodological steps applied, which enables verification by other researchers. However, it is crucial to emphasize that the practicality and effectiveness of the suggested methodology have been assessed through a case study, wherein certain steps have been only partially implemented. This is due to the extensive work and time required for their full execution, exceeding the scope of this paper. Therefore, the case study relies on a limited realization of certain steps of the proposed methodology.

### 4.1. Micro-SUMP Scope

Bar is a coastal city with approximately 14,000 inhabitants, situated in the southern region of Montenegro along the southern part of the Adriatic Sea. The Port of Bar, established in 1906, serves as Montenegro's primary cargo port. This port is situated to the

west of the town of Bar. Currently, two main operators, "Port of Bar" H. Co. and "Port of Adria," handle more than 95% of the country's maritime transport. The port area covers 200 hectares, with an additional 400 hectares earmarked for future development, according to the detailed urban plan for the port area. The Port of Bar is an international commercial seaport strategically located, with excellent accessibility and proximity to the airport and the railway station. As of now, the Port of Bar's facilities are organized as depicted in Figure 4: "Port of Bar" H. Co.(Bar, Montenegro) manages the liquid cargo terminal, dry bulk cargo terminal, Ro-Ro terminal, and passenger terminal. Meanwhile, the "Port of Adria" (Bar, Montenegro) oversees the general cargo terminal, the container terminal, and the timber terminal.

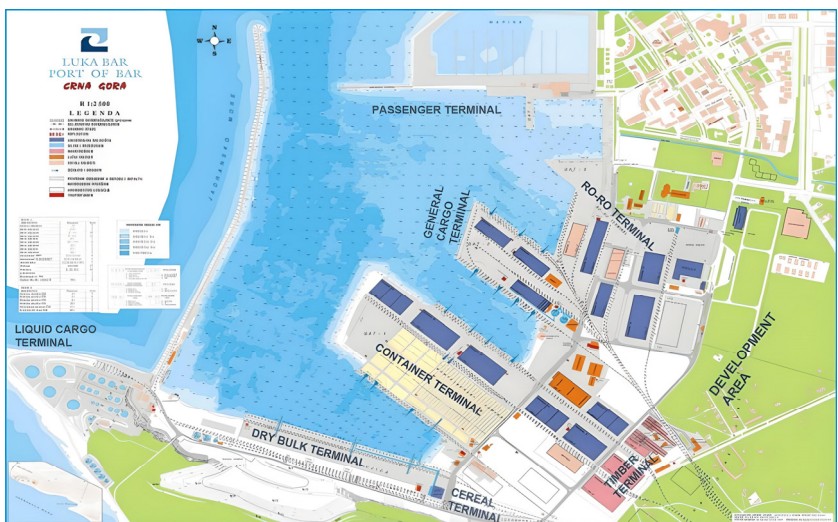

**Figure 4.** Terminals in the Port of Bar [44].

The scope of the micro-SUMP encompasses the Port of Bar and its adjacent surroundings or hinterland. The primary objective of the micro-SUMP is to facilitate the seamless and safe movement of people, primarily those arriving from the ship passenger terminal, as well as port workers and other commuters, connecting them with the surrounding areas. This plan aims to enhance the accessibility of the port area and encourage sustainable mobility options. Consequently, the micro-SUMP assesses the region around the passenger terminal, in addition to other transportation links in the vicinity of the port area.

*4.2. Step 1: Applicability of SUMP Concept—Port of Bar Sustainability Practices*

The micro-SUMP development process begins with an evaluation of the SUMP concept and its applicability to the port area. The City of Bar currently lacks a defined SUMP, which was one of the challenges in our research. In the absence of such a plan, we relied on the existing literature on the SUMP concept, which is briefly presented in the previous section. The level of detail is appropriate for the scope of this research—a case study that primarily focuses on assessing the feasibility of a potential pilot solution applicable to the observed port. The full implementation of the proposed solution would undoubtedly necessitate a more comprehensive analysis of the SUMP concept and the mandatory involvement of other stakeholders, such as the city administration, tourist organizations, cruise line operators, and city planners. In our research, the main participants included researchers/experts and representatives of the Port of Bar. We also collaborated with other experts in the field of SUMP, as well as representatives from other ports in the region, brought together within the framework of the SuMo project [45]. Due to the considerations mentioned above, this step of the proposed methodology was realized to the extent necessary for the case study. The primary goal was to understand the current sustainability efforts of the Port of Bar, their goals in this regard, and, more broadly, to determine whether the micro-SUMP concept development can be applied in the observed port.

After reviewing a dozen recent projects in which the Port of Bar participated, as well as their development plans, it can be concluded that the Port of Bar has recognized the significance of sustainable development and initiated its 'green port transformation' initiative. The overarching goal is to establish an energy-efficient and low-emission port operation to reduce costs, enhance overall efficiency, and improve environmental performance.

As a pivotal component of its comprehensive 'energy sustainability strategy', the Port of Bar has devised, in collaboration with the 'Sustainable Ports in the Adriatic-Ionian Region (SUPAIR)' project [46], an action plan for sustainable and low-carbon operations. This plan offers detailed and concrete guidelines on how to cost-effectively reduce the environmental impact of port operations. It outlines the port's most critical medium and long-term energy sustainability objectives, encompassing actions aimed at reducing energy consumption (e.g., oil, diesel, electricity, or gas) and mitigating harmful emissions. The action plan addresses energy sustainability aspects within both shipping and port operations and proposes two main categories of measures: Category 1 focuses on organizational aspects, while Category 2 centers on energy consumer clusters. In total, 65 distinct measures have been proposed. The efforts to address sustainable mobility issues within the port area represent a natural extension of the previous work aimed at improving the overall energy sustainability of the port.

### 4.3. Step 2: Analysis of Actual Mobility and Transport Environment

The second phase involved a comprehensive review of the existing traffic arrangements and mobility services within the area, taking into consideration all modes of transport. The Port of Bar is a vital hub for businesses and serves as the city's primary gateway for international shipping. Consequently, the current ship terminals are a significant source of traffic in the planning area. The traffic solutions near the terminals, especially parking facilities, have been developed incrementally, expanding away from the terminals—the absence of centralized parking facilities results in several separate paid parking areas. Free street parking is available but is subject to time restrictions.

According to the 2020 *Spatial Urban Plan of the City of Bar* [47], road traffic dominates the transportation of goods and passengers within the city. Passenger cars are the dominant mode of transport, accounting for nearly 90% of the total traffic flow on the city's road network, while bicycles make up less than 1%, and motorcycles constitute a slightly higher share, up to 3.7%. It is worth noting that there is no continuous monitoring of traffic intensity and composition within the municipality of Bar, except for the biggest road tunnel and access roads. Public transport buses provide access to the port, but their competitiveness with passenger cars in the municipality of Bar is limited. This mode of transport is primarily utilized by individuals who do not have access to a passenger car [48]. The limited development of the city's road network is a primary factor behind the inadequate growth of the public transport system. Intercity and international transport bus stops are not directly connected to the port terminals. The train and long-distance bus platforms are situated approximately 3km from the port area, requiring a 30 min walk. Areas designated for charter buses and taxis are reasonably close to the terminals.

The city lacks dedicated bike lanes or stations near the long-distance bus platforms or the port area. Street traffic within the port is unrestricted, but the insufficient number of parking spaces for both residents and tourists results in numerous irregularly parked cars that disrupt traffic flow. Walking or cycling to the port area is uncommon, especially during winter. People often walk from parking lots located several hundred meters away. Public transport usage does not adequately compensate for the limited walking and cycling activity.

### 4.4. Step 3: Defining Micro-SUMP Major Objectives

According to the developed methodology (outlined in Figure 3), the phase that follows the analysis of the current transport environment and mobility behavior is the definition of major micro-SUMP objectives and strategies. These strategies will be followed by the



identification of measures (practical solutions), as well as their further elaboration using cost-benefit analysis. This case study will emphasize cost-efficient measures (small-scale investments) over the large-scale projects typically associated with traditional transport planning efforts. For instance, cost-efficient measures like new cycle tracks or pedestrian zones can promote greater use of non-motorized transport modes. Additionally, the efficiency of public transport can be enhanced through initiatives such as, for example, dedicated bus lanes. Measure selection, as outlined in [49], is a highly critical aspect of the overall process of developing a SUMP. Mobility strategies and measures lie at the core of the planning approach for sustainable mobility.

Before commencing the measure selection process, it is essential to establish the SUMP's main vision and objectives. The formulation of the SUMP's main strategy is based on the current knowledge of sustainable mobility options, the interrelation of port transport activities and their traffic impact, as well as the current transport, mobility, and urban environment in the Port of Bar (modal share, quality of infrastructure, etc.). Based on this information, as well as based on the input provided by the port representatives regarding the scope of investments in potential solutions and their integration into the existing plans for improving the overall sustainability of the port, three different objectives for developing more sustainable mobility solutions in the port area of Bar are proposed:

- Reduce private car usage for commuter trips, particularly among employees in the port area, by promoting shared mobility and alternative means of transport;
- Promote non-motorized modes such as walking and cycling;
- Encourage the use of public transport.

*4.5. Step 4: Evaluation of Available Practical Solutions (Best Practices)*

The next step involves creating a list of potential solutions and assessing them, drawing from an examination of the best practices that were collected and analyzed earlier. To develop precise practical solutions for the given case study port, it is crucial to pinpoint and elaborate on the relevant measures and actions that have been implemented in similar cases. This step compiles well-established approaches and practices for finding new mobility concepts that bridge city and port areas. This collection of best practices is geared towards recognizing innovative and validated approaches for fostering more sustainable mobility. The identification of these diverse initiatives was carried out by thoroughly reviewing various publicly accessible knowledge repositories. Table 1 presents the solutions that have been identified as applicable to the case port under examination.

While the collected best practices do not aim to encompass the full range of measures and actions across Europe to support the implementation of SUMPs in the port area, they serve as a valuable knowledge base, addressing the issues, challenges, and adopted measures related to sustainable mobility in this specific city area. It is important to note that the comprehensive analysis of the solutions entailed a detailed description, which included identifying the initiators of the solution (government, city, company, people), the projected or actual costs of implementing the solution, the achieved results, the impact on the sustainability of the observed system, the possibility of replication, and the basic lessons learned from this solution. However, due to the extensive nature of the complete analysis, it has not been included in this paper due to scope limitations.

After evaluating the identified best practices, the next step involves compiling a list of potential solutions for the Port of Bar, considering the defined objectives for the development of more sustainable mobility solutions in the port area (already conducted in Step 3). All solutions are proposed by the authors in cooperation with the port representatives. The solutions related to the first objective will primarily focus on promoting hybrid and electric mobility, as well as services intended for car- or ride-sharing. Solutions pertaining to the second objective will concentrate on enhancing mobility in proximity to the port area, with an emphasis on cycling. The third proposed objective will prioritize the optimization of public transportation, emphasizing improved modal integration and travel management. All proposed solutions were chosen for their orientation towards increasing the availabil-

ity of sustainable mobility options in the port area, with a focus on the targeted types of mobility: the movement of commuters (workers or ferry passengers) and tourists. In addition, the proposed solutions have significant potential to align with the sustainability objectives across the wider City of Bar area. That is, these solutions are entirely in line with the proposed two-stage approach for developing a micro-SUMP. The list of proposed solutions for this case study can be found in Table 2.

**Table 1.** Overview of the identified best practices of sustainable mobility solutions in the port area (created by the authors).

| Best Practice Title | Best Practice Mission | Type of Measure, Source |
| --- | --- | --- |
| Coordinating bike lane planning | Integration of the bike lane network in the port area and connection between the existing bike lanes to have a complete network that will connect the port to the main points of the city. | Infrastructure measures [12] |
| The oblong coastal bike line | Introduction of a coastal bike line in Kotor (Montenegro). | Infrastructure measures [12] |
| Development of a public transport information system | Promoting the use of public transport by increasing its reliability and attractiveness and providing real-time information. | Regulatory and planning measures [12] |
| Redesign and upgrade of an urban axis | Redesign/upgrade of an urban axis based on the principle of SUMP with a high-participatory approach. | Infrastructure measures [12] |
| Carpooling system for port workers | Improved carpooling system in the port. | Regulatory and planning measures [12] |
| Electric car-sharing system for port employees | The company intends to make all its vehicles $CO_2$-free. For that aim, they only buy fully electric cars for their fleet, and some of them are used in car sharing by employees. | Infrastructure measures [50] |
| E-bike sharing system in a port | Implementation of the E-bike sharing system in the port area. | Planning/Infrastructure measures [12] |
| Port staff green mobility | The solution addresses the travel undertaken by staff to and from the port area. The plan is to foster the use of alternative travel modes (bikes, e-taxi, public transport) instead of private cars. | Regulatory and planning measures [51] |
| Micromobility in the port area with e-scooters and bikes | The solution targets the transportation of port employees between port premises and the transportation of cruise passengers from cruise terminals to the city of Piraeus. | Planning/Infrastructure measures [6] |
| Electric buses (eco-buses) for transportation in the port area | The solution targets the transportation of port employees between port premises and the transportation of cruise passengers from cruise terminals to the city of Piraeus. | Infrastructure measures [6] |
| Mobility points (intermodal hubs) | Establishing a mobility point to connect public transport with other sustainable mobility solutions. | Planning/Infrastructure measures [52] |
| Transport and active mobility in port areas | The solution contributes to and activates active mobility (walking and cycling) as a success factor for port areas. | Planning/Infrastructure measures [53] |
| Electrical car-sharing system | The solution is aimed at establishing e-car sharing to serve a variety of users and functions. | Planning/Infrastructure measures [54] |

**Table 2.** The proposed solutions for the port area of Bar (created by the authors).

| Solution Title | Description | Type of Solution |
|---|---|---|
| Active mobility | Creation of walking and cycling paths within the port area to promote and support non-motorized transport of employees and other commuters. | Infrastructure |
| Port of Bar internal transport by use of hybrid bus | Organization of the Port of Bar internal transport of employees within the port area by hybrid bus to reduce the number of vehicles entering the port. | Planning/Services |
| Transport of passengers from the Port of Bar to the Old City of Bar | Organization of the Port of Bar passenger transport of tourists to and from the Old City of Bar by hybrid bus to reduce traffic impact and to influence travel behavior. | Planning/Services |
| Port of Bar commute transport of employees to suburban areas by use of hybrid bus | Organization of the Port of Bar's commute transport of employees to and from suburban areas by hybrid bus to reduce traffic impacts and to influence the travel behavior of its employees and other commuters. | Planning/Services |
| Carpooling system for the staff of the port | The solution will allow port employees to share their cars. | Planning/Services |

*4.6. Step 5: Evaluation of Potential Solutions and Definition of Development Vision*

After conducting a comprehensive analysis of the current mobility challenges, needs, and objectives for establishing sustainable mobility options in the port area of Bar and completing a review of the best practices and proposition of feasible solutions for sustainable mobility choices, this step involves the evaluation and selection of the most appropriate option. Establishing suitable, achievable, and relevant solutions that align with the broader objectives of sustainability and effectiveness, both within the port area and the wide-city area will drive the formulation of an overarching micro-SUMP vision. This vision will be founded on the integration of sustainability as a core development principle, emphasizing a reduction in the environmental impact, as well as energy efficiency and the promotion of eco-friendly transport and mobility approaches.

*Solution 1: Active Mobility*—this proposed solution involves the establishment of designated walking and cycling paths within the port area, with the aim of facilitating non-motorized transportation for employees and other commuters. This initiative is considered infrastructural, as it incorporates specific infrastructure solutions to enhance safety for cyclists and raise awareness among other traffic participants. While the construction of the walking and cycling paths is designed to follow the main transport corridors within the port, the presence of a fence between the Port of Bar and the Port of Adria currently impedes the development of a unified and unobstructed walking and cycling pathway. However, with the potential future unification of these two companies (port operators), there remains a possibility that internal fences will be dismantled, thereby enabling the development of active mobility in the Port of Bar.

In this scenario, the proposed route for the active mobility path within the Port of Bar would resemble the illustration provided in Figure 5. When developing a dedicated infrastructure for active mobility in the Port of Bar, it is essential to consider the following factors: (i) implementing adequate lighting is crucial for ensuring 24 h cycling; (ii) maintaining year-round cycling requires winter services to clean surfaces for pedestrians and cyclists; (iii) specific attention must be given to intersections in port areas, considering the coexistence of rail and freight transport, emphasizing safety; (iv) wide and straight streets in port areas may encourage faster driving among motorized vehicles, even during turns, leading to potentially hazardous driving behaviors; (v) providing well-maintained

surfaces and sufficiently wide bicycle lanes with ample space for overtaking is essential; (vi) developing a clear routing system, supported by appropriate signage and marking critical areas for safety purposes, is crucial; and (vii) considering the long-term possibility of integrating cycling with other public transport modes for enhanced connectivity.

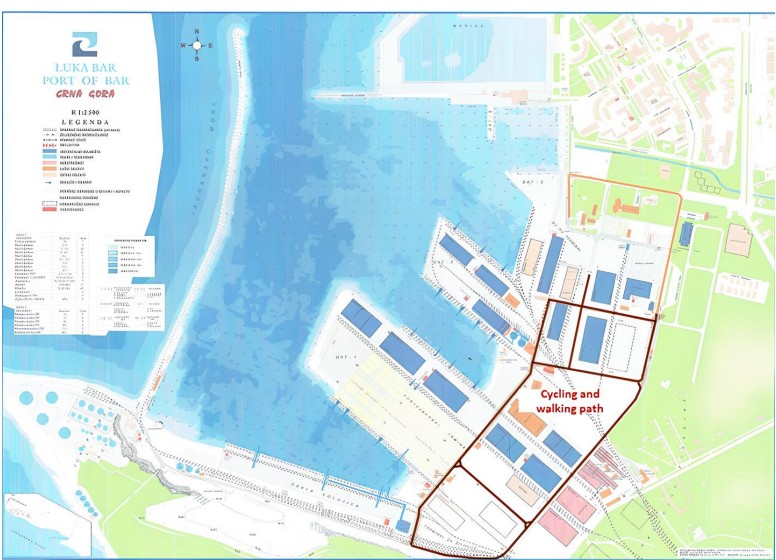

**Figure 5.** Conceptual solution of the cycling and walking path in the Port of Bar (created by the authors based on the figure from [44]).

*Solution 2: Use of hybrid bus for the Port of Bar internal transport*—this solution involves the implementation of an internal transport system within the Port of Bar, facilitating the travel of employees within the port area through the use of hybrid buses. This initiative aims to minimize the number of vehicles entering the port, thereby reducing traffic congestion, pollution, and noise within the port area. Its primary objective is to encourage the widespread adoption of low-emission vehicles for improved environmental sustainability. To assess the potential impact of the hybrid bus on internal transport within the Port of Bar, an analysis was conducted on the daily influx of vehicles entering the port, along with the number of employees working during each shift at the port. Based on the number of employees and the number of private car entries to the port area, it can be inferred that the implementation of internal transport via a hybrid bus system should accommodate approximately 300 workers during the first shift and 200 during the second shift. This capacity can be managed by scheduling an appropriate number of round trips in both shifts. The proposed route and location of the bus stops are illustrated in Figure 6.

The proposed route encompasses six stations, beginning and ending at the port's entrance gate, with four intermediate stations strategically positioned to cater to the frequent transport demands of workers during both working shifts. Implementing this solution would decrease the number of cars in the port area, positively impacting space utilization and leading to a reduction in noise and pollution levels within the port vicinity. Considering the expected peak usage of the service at the beginning and end of shifts, there would be a notable time window of 3 to 4 h during each shift where the bus could serve other purposes, such as transporting tourists between the Port of Bar and the Old City of Bar (Solution 3). This efficient utilization of capacity could yield significant financial and environmental benefits.

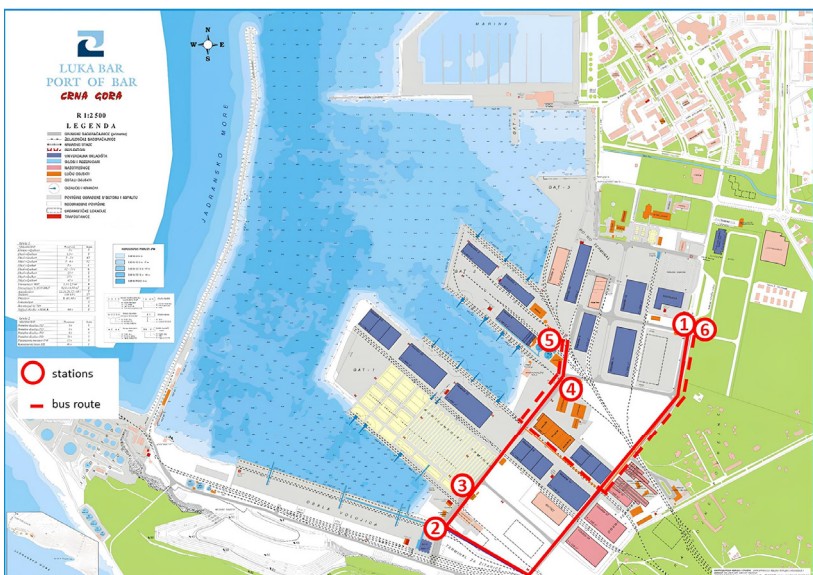

**Figure 6.** Proposed route with the location of the six bus stops (marked with numbers 1 to 6) within the port area (created by the authors based on the figure from [44]).

*Solution 3: Transport of passengers from the Port of Bar to the Old City of Bar*—this solution involves the organization of passenger transport for tourists between the Port of Bar and the Old City of Bar using hybrid buses. When analyzing passenger transport relevant to this initiative, it is essential to consider the passengers arriving from the ferry line Bar-Bari (Italy) and those arriving on cruise ships at the Port of Adria. The number of passengers passing through the passenger terminal of the Port of Bar was obtained for the period between 2011 and 2022, and the conclusion is that cruise ships bring about 15,000 passengers per year. The proposed route spans from the passenger terminal to the Old City of Bar, covering a distance of approximately 5 km. This choice is based on the Old City's significance as a key destination for tourists and considering that the city center itself is part of the pedestrian zone in proximity to the terminal (approximately 400 m).

Considering the passenger count, the capacity of the hybrid bus, and the distance of the proposed route, it is evident that a single bus will not suffice to provide the service independently. Therefore, it should be offered as a complement to an existing tourist transport service. Additionally, purchasing a larger fleet of hybrid buses to accommodate the occasional transportation for 1000 cruise tourists would not be financially justified due to the low average level of capacity utilization, considering the intermittent nature of cruise operations, which do not operate on a daily basis. As a result, the initial approach to the passenger transport of tourists between the Port of Bar and the Old City of Bar by hybrid bus should be introduced as an occasional supplementary service. This service should be prearranged in advance between the Port of Bar and the cruise tour operator, primarily during the times when the bus is not in use for Solution 2.

*Solution 4: Port of Bar commute transport of employees to suburban areas by use of hybrid bus*—this solution entails the organization of commuter transport for employees to and from suburban areas of the Port of Bar using hybrid buses. This initiative aims to mitigate traffic impacts and influence the travel behavior of both employees and other commuters. To assess the current necessity for such a service, an analysis was conducted on the number of employees commuting to the port for work from distances exceeding 5 km. Presently, there are 47 employees working in the Port of Bar who travel to work from distances greater than 5 km. However, the number of employees commuting to these specific destinations is relatively low, with less than five employees for each location. Additionally, these destinations are geographically dispersed, rendering the service financially and environmentally unviable at present. This is due to the considerable length of the route, exceeding 40 km

per direction, resulting in extended travel times and a high energy consumption rate per passenger kilometer, primarily attributed to the low-capacity utilization.

Even if the line length were to be reduced, it would have a favorable impact on the energy consumption rate. However, it would only cater to a small portion of employee commute transport, with the remaining employees likely to continue using private cars, thereby casting doubt on the overall financial and environmental efficacy of the service. Therefore, initially, it would be more effective to postpone the implementation of this service and instead focus on resolving this issue through the introduction of a carpooling system (Solution 5). Subsequently, this service should be considered for future implementation in collaboration with the Port of Adria and the City of Bar public transport company to address other city commute transport demands and enhance capacity utilization, thereby augmenting its financial and environmental impact. In the event that this service is realized, it is imperative to note that workers would need to disembark the bus before the port's main gate for security checks and then re-board the bus within the port area in accordance with safety regulations.

*Solution 5: Carpooling system for the staff of the port*—the primary objective of this solution is to encourage the efficient utilization of private cars among port workers, leading to a reduction in emissions and noise disturbance resulting from commuting to and from the port area. This initiative aims to enhance the public space within the port area and generate savings on fuel costs. Supporting carpooling by the port, particularly during periods of high pollution or elevated fuel prices, presents an effective approach to maximize the seating capacity of a car that would otherwise remain unused if only the driver were to use the vehicle. By accommodating multiple passengers in one vehicle, carpooling reduces individual travel expenses, including fuel costs and tolls, while alleviating the stress associated with driving. Moreover, carpooling represents a more environmentally friendly and sustainable mode of travel, as the sharing of journeys contributes to a reduction in air pollution, carbon emissions, traffic congestion on roads, and the demand for parking spaces.

In the context of the Port of Bar, the promotion of carpooling should involve the development, procurement, or lease of an application designed to facilitate ride-sharing among two or more employees. This application would encourage carpooling by analyzing factors such as employees' shifts, working hours, locations, routes, and other relevant criteria. By mapping nearby colleagues geographically, the application would effectively assist in establishing carpools for commuting to and from work. The application can be developed and maintained by the port, or alternatively, it could be leased from any of the available online carpooling app providers and offered to employees at no cost. The price for leasing such an online application for companies ranges typically from 350 to 1500 EUR per month, depending on the app's features and the number of users within the company. Implementation of the application can be completed within one or two workdays. Once the application is developed or leased by the Port of Bar, it should be actively promoted among employees to encourage its use and ensure its positive impact on the environment. Initially, the port should provide support or incentives to employees to encourage the uptake of the application.

### 4.7. Step 6: Selection of Solution and Verification of Its Feasibility for Implementation

The final step assumes the selection of an appropriate solution based on the previously conducted detailed description and qualitative analysis, along with the identification of potential specific issues related to its implementation. The validation of the chosen solution's application is assessed through the use of cost-benefit analysis (CBA).

After careful consideration of the proposed solutions detailed in the previous step, it was determined that the CBA would only be conducted for Solution 2. This decision was reached based on several factors. The implementation of Solution 1 would necessitate the integration of two operators (the Port of Bar and the Port of Adria), which may or may not occur within a defined time frame. Implementing Solution 3 would require close collaboration with the city's tourist organization and the cruise line operator to

accurately project service demand and plan the necessary capacities accordingly. Similarly, the implementation of Solution 4 would entail collaboration with the City of Bar and its public transport company, necessitating a more extensive analysis involving data that is currently unavailable. Additionally, conducting a CBA for the proposed Solution 5 would require a precise definition of the carpooling application architecture, prompting inquiries from IT companies for financial proposals for application development. This process would demand a significant amount of work and time, which is currently not feasible within the scope of this case study. In the end, Solution 2 adhered to the principle of integrated sustainable development for both the City of Bar and the Port of Bar. This encompassed considerations related to the current traffic situation and traffic management within the City of Bar, its spatial planning, and the sustainable development plans for the Port of Bar area. Finally, according to [55], this solution of the organization of bus services is the most typical mobility management initiative by port companies.

In accordance with common guidelines [56], an appropriate methodology was employed to conduct a rational and sustainable CBA. The first step involved identifying critical issues pertaining to the implementation of the selected solution. The second step entailed specifying the characteristics of the proposed hybrid bus and the bus line for the pilot action. Detailed CBA calculations are provided in Appendix A. The analyzed pilot project, which represents a scenario with a diesel hybrid bus, was tested and compared with the current situation where all the internal mobility within the port area is carried out by private cars. Considering the estimated investment cost, benefits for both transport users and non-users, as well as the external costs saved, a CBA was performed to verify the economic convenience of the proposed solution. The comparison between the two alternatives was conducted by analyzing the difference between the costs and benefits over the years.

To assess the feasibility of the solution (pilot project), the following indicators were utilized: the net present value (NPV) of the project, calculated using a discount rate of 5% [57], and the pay-back period (PBP). The determination of the NPV and PBP is based on the financial cash flow, and the financial calculations are elaborated in Appendix A. Table 3 and Figure 7 compare the results of the CBA conducted in two ways: the "traditional" approach (without considering the external costs) and the approach that accounts for the overall carbon footprint. The table shows that the pay-back period for the "traditional" CBA is 17 years, while for the CBA, based on the carbon footprint, it is 11 years.

**Table 3.** Results of CBA for the selected solution (created by the authors).

| CBA Indicators | "Traditional" CBA | CBA with the Carbon Footprint |
|---|---|---|
| Pay-back period (years) | 17 | 11 |
| NPV in 10 years (in EUR) | −83.6855 | −19.7264 |
| NPV in 15 years (in EUR) | −13.7019 | 72.2728 |
| NPV in 20 years (in EUR) | 41.132 | 144.356 |

As evident from Table 3 and Figure 7, a comparison of the cost-benefit indicators reveals that the analysis based on the overall carbon footprint resulting from the pilot action provides a better estimation of the positive impact of using hybrid buses. Since hybrid buses have a lower carbon footprint compared to traditional diesel vehicles, transportation services based on this technology offer clear benefits in comparison to the current situation. While the investment costs for hybrid buses are relatively high, the financial effects, in terms of the pay-back period, may not be as favorable when only direct transportation costs are considered (the "traditional" way). However, when both direct and indirect (external) transportation costs are factored in, along with the potential to repurpose the port area currently used for car parking into a profitable activity, the financial effects could become significantly more favorable. A detailed overview of the expected effects of implementing the pilot action (selected solution) is provided in Appendix A.

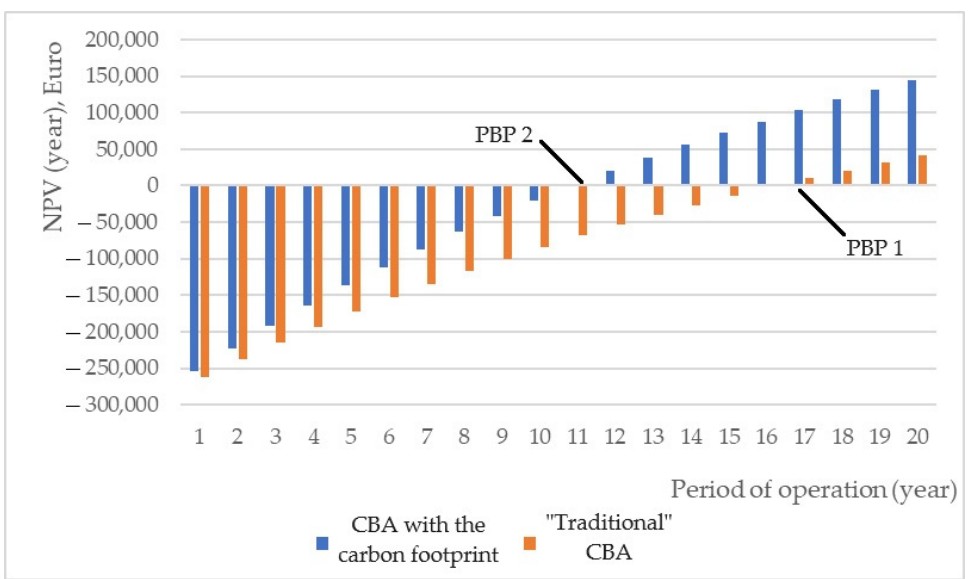

**Figure 7.** Dependence on NPV of investment in the pilot project (created by the authors).

Given that the primary focus of the pilot action is expected to be on environmental and social benefits rather than financial gains, an additional SWOT analysis of the hybrid bus service has been conducted (Table 4). Integrating the results of a CBA with a SWOT analysis can offer a more comprehensive understanding of the pilot study's feasibility. The primary strengths identified, including better space utilization (linked to the existing parking space), no significant infrastructure investment, and the relatively low resource requirements for implementation align well with the project's fast PBP. On the other hand, the CBA overlooks the additional planning and organization required, as well as the additional maintenance costs related to hybrid bus implementation, which represents the project's weaknesses. Therefore, in order to successfully implement the proposed solutions, the Port of Bar must address the following issues: acquiring new tools and spare parts for maintaining the hybrid powertrain and electrical subsystems, managing the upkeep of the newly created bus stops and the depot, etc. A significant weakness lies in the inefficient use of bus capacity when the service is exclusively oriented towards commuters. Hence, the port authorities would benefit from integrating this service with tourist transportation from cruise ships, fostering a stronger relationship with both the city's tourist organization and the cruise line operator.

The potential integration of this service into urban public transport, along with its compatibility with other sustainable mobility options like car sharing, enhances the feasibility of implementing this solution. Moreover, the substantial interest in optimizing the use of the port area's surface, currently designated for vehicle parking, not only aligns with the results of the cost-benefit analysis but also serves as a significant incentive to actualize this solution. The CBA indicates that the deployment of hybrid buses could generate profit for the port authorities, creating an opportunity for a new business model related to the rental of parking spaces. The potential threats identified, such as the initial resistance of workers to using this service, could impact the benefits outlined in the CBA. Additionally, threats related to the reduced attractiveness of the proposed solution, both conceptually and technologically, could affect the potential estimated benefits of its implementation.

To sum up, following the proposed methodology and implementing all the necessary steps, a sustainable mobility solution has been identified as the foundation for the future micro-SUMP for the Port area of Bar. The key strategic element of this micro-SUMP is the introduction of a hybrid bus for internal trips within the port area. The proposed solution is in accordance with the integrated approach in the development of the micro-SUMP because it is highly suitable for integrating into the development of city-wide SUMP for the City of Bar.

**Table 4.** SWOT analysis of the pilot action implementation (created by the authors).

| Strengths | Weaknesses |
|---|---|
| Better space utilization. Reduction in noise level. Decreased air pollution. Improved safety conditions and security procedures (reduction of approximately 30–50% in time for security procedures at the port gate). No significant infrastructure investment. Relatively low resource requirement for implementation. | Inefficient use of bus capacity occurs when only this type of service is used (involving eight round trips, which amounts to approximately 4 h of active transport time, resulting in utilization below 20%). Financially unprofitable, especially when considering only direct operating costs, resulting in a low pay-back period. Additional planning and organization are required. Additional maintenance costs. |
| **Opportunities** | **Threats** |
| Enhancement of the team spirit among workers. Introduction of additional sustainable transport services both within and outside the port area. Integration of the service with other environmentally sustainable public transport services. Compatibility with other potential environmentally sustainable services, such as car sharing for commuting and green mobility. Transformation of the port area, currently used for car parking, into a business purpose that can generate income. | Initial resistance to changes by the port workers. Lack of general interest from city authorities to integrate this solution into sustainable mobility plans at the city-wide level. It can become an isolated solution without integration with future micro-SUMP solutions. Moving beyond hybrid buses in favor of alternatives, such as fully electric buses or hydrogen buses. |

## 5. Discussion

The contribution of the research results is two-fold: at the theoretical level, it provides a methodology for the development of SUMP for individual micro-territorial entities, and on the practical level, it demonstrates the real-world value of this methodology through the successful implementation in a case study. The developed methodology aligns closely with the European recommendations and SUMP development guidelines, although this alignment is not always observed in European studies of SUMP development [58]. The results of our paper affirm the adaptability of mainstream European SUMP development guidelines to the context and specific requirements of urban areas and territories in which SUMP is applied. The methodology devised for creating a micro-SUMP tailored to the port area corresponds to general SUMP methodology in its recommendations concerning stakeholders, main phases in the analysis, integration of transportation modes, measured planning and analysis, measurability of targets, integration of costs, etc. The provided methodology addresses the challenge of developing a micro-SUMP for specific parts of the urban areas, a topic seldom covered in the existing literature. The case study results confirm that the effectiveness of the developed methodology can be assessed based on the quality, relevance, and replicability of the proposed, locally relevant actions designed to achieve the objectives of the specific territory, akin to the approach in [59]. Naturally, the methodology also ensures the smooth implementation of the obtained results into a city-wide SUMP development.

The developed methodology shows significant robustness, as its practical utility has been validated using a real case study example. Namely, the case study of the Port of Bar demonstrates the methodology's applicability and how it addresses the challenges and goals of sustainable transport planning in port areas. This gives us confidence in anticipating that this methodology can be used with high accuracy in other port areas as well. However, the practicality and efficiency of the proposed methodology have been evaluated via a conducted case study, albeit with partial implementation of certain

methodology steps. The decision to undertake only a partial implementation was driven by the substantial work and time demands exceeding the scope of this paper. Consequently, it is crucial to emphasize that the case study is based on a limited realization of certain steps outlined in the proposed methodology. This also constitutes the initial constraint of the conducted research.

Transport planning within port areas is a complex task and requires the involvement of a number of stakeholders. Although the methodology of this research assumes that all relevant stakeholders should be included in the decision-making process of the micro-SUMP solution, they did not all participate in the case study. Concerning stakeholder involvement, apart from port authorities and experts from the field of SUMP, other stakeholders, such as municipal authorities and urban city planners, tourist and cruise line operators, local communities, and city public transport companies, should be included. The primary reason for their absence is the low level of interest in addressing this issue within the port area. This represents another significant limitation of the research, as the absence of relevant stakeholders may potentially impact the quality of decisions of a well-defined micro-SUMP development framework. Hence, executing the developed methodology in detail would require a more robust collaboration with all stakeholders. This can be attained by establishing a collective understanding of their roles and opportunities early on and by disseminating among them the micro-SUMP mission, vision, methodology, objectives, and actions. Such an approach would guarantee their involvement, active engagement, and consensus in the decision-making process. Facilitating such a collaboration can be achieved through organizing events such as mobility forums, round tables, and public debates. The research was also limited by the case study itself. Although the methodology implies the development of a micro-SUMP involving the creation of a fundamental SUMP, developing a solution for the Port of Bar fell short of that need due to the absence of a fully established SUMP for the entire city.

On the other hand, the developed methodology and its application in the case study for the Port of Bar are in strong alignment with the existing city transportation policies and regulations. It aligns completely with the city's aspirations for sustainable transportation development and can serve as a good pilot for defining a similar sustainable mobility solution at the city-wide level. Since the scalability of the project is very good, it allows the solution from the case study to be scaled to an entire fleet of hybrid buses that will cover the whole City of Bar area. Hence, the lessons learned during the implementation of the micro-SUMP methodology suggest that the development of city-wide SUMP and micro-SUMP could occur bi-directionally or circularly, where the development of micro-SUMP solutions can initiate and encourage the development of a city-wide SUMP. This finding represents an important contribution to existing research, as the possibility of the stated two-way relationship was not previously taken into account. Also, this finding can be viewed as a proposal for the potential improvement of the developed methodology for its future applications.

The final limitation of the conducted case study is the absence of a specific strategy for the 'step-by-step' implementation of the proposed solution. This strategy should be directly related to the budget definition, necessary financial resources, and possible sources of financing. In essence, this research does not provide a plan for the operational implementation of the proposed solution, nor does it include an implementation schedule that clarifies which entities will take responsibility for various activities. Moreover, it does not specify the resources, institutional support, and legal approvals needed to carry out these activities effectively.

The advantage of the developed methodology lies in its recognition that the development of the micro-SUMP for port areas must align not only with city-wide sustainable mobility planning but also with regional collaboration plans, given that the port's transportation and mobility system is connected to the broader urban system and extends into regional and cross-border areas. This consideration significantly influenced the selection of a specific solution in the case study, given its high scalability and transferability capacity

to other port cities. This was confirmed by the partner ports within the aforementioned SuMo project [45]. The proposed case study's solution can contribute to the sustainability of the region because the connection with other ports in the region through ferries becomes more sustainable as the first and last legs of the journey—which are realized on the mainland—use a sustainable mode of transport.

Finally, the implemented methodology and the solution deployed in the Port of Bar as part of the micro-SUMP development, according to the CBA and SWOT analysis, will have a wide range of environmental and social impacts, including a reduction in emissions, improved air quality, reduced congestion in the port area, and enhanced sustainable mobility for commuters. The suggested methodology has the potential for future refinement to create and formulate optimal micro-SUMP solutions for any given port city. To assess and select alternative sustainable mobility solutions, a multi-criteria analysis involving all relevant stakeholders can be conducted. Additionally, a promising avenue for future research could involve comparing public perceptions, port expert opinions, and choices made by city mobility experts when identifying the most appropriate mobility solutions for a particular port area.

## 6. Conclusions

The present research aims to contribute to the field of development and implementation of sustainable urban mobility plans (SUMPs) by proposing a methodology tailored to create a territorial SUMP that addresses the specific needs and circumstances of a port and its connected urban regions. To achieve this objective, in the initial phase, the paper outlines an approach and methodology, which encompass several steps essential for constructing a comprehensive territorial or micro-SUMP. This approach places particular emphasis on adopting an integrated perspective that considers the sustainable development of both the urban city area and the port area, acknowledging their interconnectedness. In the second phase, the methodology is put into practice through a real case study—the Port of Bar. In this application, the methodology steps are systematically followed, leading to the practical implementation of the chosen mobility solution: the use of a hybrid bus for internal employee transport within the port area. The conducted research has some limitations, primarily related to the case study realization. Firstly, the case study assumed a limited implementation of certain steps of the proposed methodology. Secondly, although the methodology presumes the involvement of all relevant stakeholders in the decision-making process for micro-SUMP solutions, not all participated in the study. Thirdly, the case study did not include a plan for the operational implementation of the proposed solutions. These limitations are attributed to the substantial work and time required to overcome them, which significantly exceeds the scope of this research. Despite these shortcomings in the realized case study, it can be asserted that it demonstrates the ease of use of the methodology and its high practical applicability and adaptability.

The proposed methodology can support port authorities and, more generally, the public and private decision-makers involved in the development process of port sustainability. Both the methodology and the findings from the case study can be utilized in similar ports in the region and beyond. In other words, the obtained results demonstrate a high level of transferability, a fact corroborated by other ports in the area. To sum up, this paper contributes to the promotion of SUMPs for improving port sustainability, provides valuable insights and concrete solution developments on how to plan sustainable mobility solutions for both port areas and city-wide areas, and promotes alternative means of transport and public transport. In addition, the presented case study also provides results of the cost-benefit analysis (CBA) that could be applied, at least to some extent, in decision-making for various types of mobility solutions. Future research and development related to micro-SUMP methodologies and their implementation will remain a goal for years to come, given that working towards sustainable and low-carbon mobility will be the focus of both EU and national policies. This ensures that the relevance of the subject discussed in this paper will remain very high in the foreseeable future.

**Author Contributions:** Conceptualization, M.M., S.B. and R.M.T.; methodology, M.M. and S.B.; validation, D.M., S.N. and R.M.T.; formal analysis, M.M. and R.M.T.; data curation, S.B. and S.N.; writing—original draft preparation, M.M. and S.B.; writing—review and editing, R.M.T. and D.M.; visualization, D.M.; supervision, M.M. and S.N. All authors have read and agreed to the published version of the manuscript.

**Funding:** This research received no external funding.

**Institutional Review Board Statement:** Not applicable.

**Informed Consent Statement:** Not applicable.

**Data Availability Statement:** Data are contained within the article.

**Acknowledgments:** The realization of this paper has been supported by the Serbian Ministry of Education, Science, and Technological Development program through project no. 451-03-68/2020-14/200156: "Innovative scientific and artistic research from the FTS (activity) domain". The authors also wish to acknowledge the contribution of the SuMo project (Sustainable Mobility in the Port Cities of the Southern Adriatic Area), co-financed through the Interreg-IPA CBC Italy–Albania–Montenegro program. Special thanks are extended to the SuMo project partners from the Port of Bar for their cooperation within the framework of this project and case study, which has yielded important results for furthering the implementation of the measures aligned with the sustainable mobility principles.

**Conflicts of Interest:** The authors declare no conflicts of interest.

## Appendix A. Cost-Benefit Analysis for Implementation of the Selected Solution

A cost-benefit analysis (CBA) is a systematic method for quantifying, assessing, and contrasting the advantages and drawbacks associated with decisions, actions, or plans. It revolves around the assignment of monetary values to all relevant activities. Consequently, if the anticipated benefits outweigh the expected costs, it is advisable to proceed with the decision, action, or plan. CBA is widely applied to gauge the viability and profitability of public policy interventions by comparing the overall investment needed to execute a given strategy or project with its potential returns. Typically, CBA methodologies emphasize economic considerations, often incorporating metrics like pay-back period (PBP) and net present value (NPV).

**Basic inputs**

The first activity of the given CBA analysis was the identification of the main elements characterizing the implementation of the proposed solution: hybrid bus transport service quality, travel time per period of the day, etc. The second activity was related to specifying the characteristics of the proposed hybrid bus and bus line, such as the length, number of stops, frequency, etc. Therefore, the main characteristics of implementation, or the basic inputs for the CBA are: one new hybrid diesel-electric bus; 3.3 km length for the proposed bus line; six stops on the bus route; 25–30 min average turnaround time of the bus route; a design frequency of eight round trips per day (a function of the demand estimated for this proposed line); a maximum route capacity of 100 passengers per direction, or a maximum of 200 passengers per the round route; an estimated average fuel consumption of the hybrid bus—in the regime of the slow motion (considering the short distances between the stations) is 35 L/100 km, i.e., 1.2 L/round trip; an average $CO_2$ emission of the hybrid bus—in the regime of the slow motion (considering the short distances between the stations) is 2.1 kg/km; an estimated average $CO_2$ emission of the private cars (13 years old on average)—in the regime of the slow motion is 160 g/km; an operation time for the new bus of 8 h/day; the yearly operation time is 300 days/year; the average number of private cars currently entering the port area is 390 per day; the average number of private cars entering the port area in the proposed scenario is 30 per day (explanation: there are currently 640 permits issued for the private car entries to the port area and after the full pilot action implementation, around 590 permits should be canceled (92.2%). Considering the current situation with approximately 390 cars per day entering the parking locations in the port and the permit reduction of 92.2%, the approximated reduction of private car

entries will be around 360 per day, which means 30 car entries per day in the scenario with the diesel hybrid bus); the average distance traveled by private cars within the port area is 1.1 km; the average private car fuel consumption is 0/08 L/km; and the average hybrid bus fuel consumption is 0.35 L/km.

Estimation of investment cost

The assessment of the economic returns on the investment was initiated using the unit value, as presented in Table A1.

**Table A1.** The cost unit value.

| No. | Unit | Value |
|-----|------|-------|
| 1 | Diesel prices (EUR/L) | 1.48 |
| 2 | Acquisition cost for one diesel hybrid bus (EUR) | 277,000.00 |
| 3 | Average hybrid bus consumption (km/L) | 2.8 |
| 4 | Costs for bus stop development (EUR) | 10,000.00 |
| 5 | Costs for drivers' salary | n/a * |

* The company will reorganize job positions instead of hiring a new person.

The benefits estimation

The benefits, which represent the positive impacts resulting from the proposed bus service, were estimated by comparing the conditions of the pilot project implemented to those without it. These estimated impacts include: (i) benefits for transport users, encompassing both users of the proposed bus line and other members of the transportation system within the port area (such as private car and bus users who benefit from reduced port entrance congestion and increased parking space availability); and (ii) benefits for non-users, referring to those who do not use the proposed bus service but still experience positive impacts, such as reductions in pollutants and an improvement in the quality of life within the port area.

Regarding the estimation of benefits for the users, the proposed bus line will generate two types of benefits: (i) those directly perceived by the port authorities (e.g., reduced costs for parking space) and (ii) those directly perceived by the workers (e.g., reduced costs for car use and maintenance). A significant portion of the cost-benefit evaluation pertains to estimating the external impacts resulting from the project, often referred to as externalities, which affect both the environment (e.g., climate change costs) and human health (e.g., air pollution and noise). The introduction of the new bus service is expected to lead to a reduction in car usage, generating external benefits for non-users. The external impacts or benefits that were estimated include variations in terms of climate change, air pollution, and noise. To estimate the monetary value of these benefits, the estimated changes in pollutant emissions and consumption emissions were multiplied by the marginal costs. The marginal costs used for this calculation were based on those proposed by the European Commission, utilizing the COPERT model [60]. Table A2 summarizes these benefits.

**Table A2.** The estimated benefits.

| Type of Benefits | Benefits for the Users | Benefits for the Non-Users |
|------------------|------------------------|----------------------------|
| [B1] Saving costs for charging parking spaces (EUR/year) | 16,500 | - |
| [B2] Saving costs for fuel consumption (EUR/year) | 9830.16 | - |
| Total $CO_2$ emission reduction (kg/year) | - | 2376 |
| [B3] External cost saved (EUR/year) | - | 8283 |

NPV and PBP calculation

The net present value (NPV) is a measure of the profitability of an investment calculated by subtracting the present values of cash outflows, which include the initial cost, from the present values of cash inflows over a specified time period: $NPV(r) = \sum_{t=0}^{T} \frac{\sum_j B_j^t}{(1+r)^t} - \sum_j C_j^t$, where r is the rate of return (equal to 5%, as proposed in [57]); T is the time period equal to 10/15/20 years; $B_j$ is all the benefits (both for the users and for the non-users) that the new bus service will produce; $C_j$ is all the costs supported (investment). The pay-back period (PBP) is the period of time required to recoup the funds expended in an investment or to reach the break-even point (return of the investment): $PBP = T_{min}; NPV(r) > 0$.

In the given case study, the benefits and investment are as follows: $B_1$ = 16.500 EUR per year (300 days $\times$ 55 EUR/day); $B_2$ = 9830.16 EUR per year; $B_3$ = 8283 EUR per year; and $C_1$ = 277,000 + 10,000 = 287,000 EUR—the acquisition cost for the diesel hybrid bus and bus stops development. The CBA was conducted in two ways: the "traditional" approach (without considering the external costs) and the approach that accounts for the overall carbon footprint. The pay-back period for the "traditional" CBA is 17 years, while for the CBA, based on the carbon footprint, is 11 years. An overview of the expected effects of implementing the pilot action (selected solution) is given in Table A3 below.

**Table A3.** Overview of the solution implementation's expected effects.

| Comparison | Current Scenario: 390 Cars/Day Entering the Port Area | Implemented Solution: 30 Car Entries and 8 Round Trips/Day with 1 Hybrid bus |
|---|---|---|
| Total distance traveled (km/day) | 429 | 59.4 |
| Average fuel consumption (L/km) | 0.08 | 0.35 for buses and 0.08 for cars |
| Total fuel consumption (L/day) | 34.3 | 11.9 |
| Total quantity of $CO_2$ (kg/day) | 68.7 | 60.7 |
| Total direct transport costs (EUR/day) | 50.8 | 17.6 |
| Total indirect (external) transport costs (EUR/day) | 32.05 | 4.44 |
| Total amount of dedicated parking space ($m^2$) | 6500–7000 | 1000–1200 |

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
