# Peer review of "Sustainable Urban Mobility Planning in the Port Areas: A Case Study"

_sustainability, doi:10.3390/su16020514_

Round 1

Reviewer 1 Report

Comments and Suggestions for Authors

The article provides a comprehensive exploration of sustainable urban mobility planning in port areas, focusing on the development and application of a tailored methodology. While the overall structure is sound, there are opportunities for improvement, as highlighted in the provided feedback.

  1. Section 2 - Theoretical Framework: The theoretical framework in Section 2 appears overly detailed. Streamlining this section without compromising essential information would enhance the overall flow and reader engagement. Ensure that each detail directly contributes to the core concepts and objectives of the study.

  2. Case Study Analysis: The case study analysis is commendable for its thoroughness, encompassing various alternatives and considering causes and effects. However, to augment the study's robustness, incorporating perspectives from key stakeholders, such as port workers, would provide valuable insights into the practicality and acceptance of proposed mobility solutions.

  3. Cost-Benefit Analysis: The detailed analysis of the cost-benefit aspect, while commendable, could benefit from graphical representation for clarity. Introducing visual elements would enhance the reader's understanding of the factors involved. Additionally, linking the SWOT analysis findings more explicitly to the cost-benefit analysis would strengthen the discussion on project risks and potential mitigations.

  4. SWOT Analysis: The SWOT analysis is insightful, yet the section on threats could be expanded. If there is only one threat identified, it is crucial to delve deeper into its significance and potential implications. This would provide a more nuanced understanding of the risks associated with the project and allow for a more comprehensive risk management strategy.

  5. Conclusion: The conclusion effectively summarizes the study's objectives, methodology, and findings. To enhance this section further, consider emphasizing the specific contributions of the research to the existing literature and reiterating the real-world applicability of the proposed methodology.

Reviewer 2 Report

Comments and Suggestions for Authors

the manuscript has numerous typos and grammatical errors . It is necessary to better emphasise the novelty of the research in the introductory part

all images should contain the source and reference link if not made by the author

More bibiliographic references regarding the holistic approach of SUMPs should be included

Therefore we recommend reading 

1)Miltiadou, M., Taxiltaris, C., Mintsis, G., Basbas, S., Tsoukala, A., Fylaktakis, A., & Panousi, E. (2021). Case Studies of Sustainable Urban Mobility Plans and Measures Implemented in the Framework of the Sumport Project. WIT Transactions on The Built Environment204, 259-273.

2) Charradi, I., Campisi, T., Tesoriere, G., & Abdallah, K. B. (2022, July). A Holistic Approach to SUMP Strategies and Actions in the Post-pandemic and Energy Crisis Era. In International Conference on Computational Science and Its Applications (pp. 345-359). Cham: Springer International Publishing.

What are the limitations of this research?

Is there replicability in the study in other European and non-European contexts?

A paragraph with conclusions is missing, which does not yet contain tables but a critical view of what has been written 

Comments on the Quality of English Language

Moderate editing of English language required

Author Response

To:   Anonymous Reviewer 2

REVISED VERSION

Sustainable Urban Mobility Planning in the Port Areas: A Case Study

Reviewer #2:

Comments to the Author

Comment 1: The manuscript has numerous typos and grammatical errors . It is necessary to better emphasise the novelty of the research in the introductory part.

Answer: Thank you very much for sharing your comments and suggestions with us. They are a very useful guide for improving our work. We think that they enhance the quality of our research and hope that we have fulfilled all your requests according to your wish. Moreover, revised manuscript has now been proofread by a native competent English professor and all reviewer suggestions have been taken into account. Additionally, in the introductory chapter, text has been added to better emphasize the significance and novelty of the paper (added text from line 70 to line 85 in the Track Changes version of the paper).

Comment 2: All images should contain the source and reference link if not made by the author.

Answer: Thank you very much for the comment. All images have been properly referenced, and the remaining ones are labeled as the work of the paper's authors.

Comment 3: More bibliographic references regarding the holistic approach of SUMPs should be included. Therefore, we recommend reading:

  • Miltiadou, M., Taxiltaris, C., Mintsis, G., Basbas, S., Tsoukala, A., Fylaktakis, A., & Panousi, E. (2021). Case Studies of Sustainable Urban Mobility Plans and Measures Implemented in the Framework of the Sumport Project. WIT Transactions on The Built Environment, 204, 259-273.

Answer: Thank you for your comment. This reference has already been included in the paper (listed as number 6 in the initially submitted version of the paper).

  • Charradi, I., Campisi, T., Tesoriere, G., & Abdallah, K. B. (2022, July). A Holistic Approach to SUMP Strategies and Actions in the Post-pandemic and Energy Crisis Era. In International Conference on Computational Science and Its Applications (pp. 345-359). Cham: Springer International Publishing.

Answer: This reference has been added, along with the following references: 16, 17, 18, 19, 26, 27, 28, and 31. These additional references provide a more detailed explanation of the concepts of smart and sustainable mobility, as well as the SUMP concept and its application in various conditions. In total, 23 new references have been added (a complete list is provided at the end of the response letter).

Comment 4: What are the limitations of this research?

Answer: Thank you for your valuable comment. In the revised Chapter 5, "Discussion," several limitations of the conducted research are specified (lines 861-877, and lines 905-912 in the Track Changes version of the paper).

Comment 5: Is there replicability in the study in other European and non-European contexts?

Answer: Many thanks, this comment has been addressed throughout the Chapter 5. In the modified Chapter 5, "Discussion", characteristics of the transferability of the proposed solution to other European ports are outlined in the text from line 913 to line 921 (in the Track Changes version). This aspect is also emphasized in the Conclusion section, specifically in lines 970-973 (in the Track Changes version).

Comment 6: A paragraph with conclusions is missing, which does not yet contain tables but a critical view of what has been written.

Answer: Thank you. The "Discussion" and "Conclusions" chapters have been revised and improved, with a particular focus on the "Discussion" chapter where a critical view of the written content has been provided.

Comment 7 (Comments on the Quality of English Language): Moderate editing of English language required.

Answer: Thank you for a comment. The English language is now modified and checked by native competent English professor.

We hope that all the amendments discussed above meet all your concerns with regards to the content of this work and its presentation.

List of added references

  • European Commission. Communication from the Commission to the European Parliament, the Council, the European Economic and Social Committee and the Committee of the Regions-Together towards competitive and resource-efficient urban mobility. White paper {SWD(2013) 524 final-529 final}, Brussels 2013. [https://eur-lex.europa.eu/resource.html?uri=cellar:82155e82-67ca-11e3-a7e4-01aa75ed71a1.0011.02/DOC_3&format=PDF]
  • Lozano, D.L.A; Marquez, S.E.D.; Puentes, M.E.M. Sustainable and smart mobility evaluation since citizen participation in responsive cities. Transportation Research Procedia 2021, 58, 519-526 [https://doi.org/10.1016/j.trpro.2021.11.069]
  • Benevolo, C.; Dameri, R.P.; D’Auria, B. Smart mobility in smart city. In Empowering Organizations, Lecture Notes in Information Systems and Organisation; Torre, T.; Braccini, A.; Spinelli, R. Eds.; Springer, 2016, 11, 13-28 [https://doi.org/10.1007/978-3-319-23784-8_2]
  • Lyons, G. Getting smart about urban mobility – Aligning the paradigms of smart and sustainable. Transportation Research Part A 2018, 115, 4-14 [https://doi.org/10.1016/j.tra.2016.12.001]
  • Melkonyan, A.; Gruchmann, T.; Lohmar, F.; Bleischwitz, R. Decision support for sustainable urban mobility: A case study of the Rhine-Ruhr area. Sustainable Cities and Society 2022, 80, 103806 [https://doi.org/10.1016/j.scs.2022.103806]
  • Mozos-Blanco, M.A.; Pozo-Menendez, E.; Arce-Ruiz, R.; Baucells-Aleta, N. The way to sustainable mobility. A comparative analysis of sustainable mobility plans in Spain. Transport Policy 2018, 72, 45-54 [https://doi.org/10.1016/j.tranpol.2018.07.001]
  • Charradi, I.; Campisi, T.; Tesoriere, G.; Abdallah, K.B. A Holistic Approach to SUMP Strategies and Actions in the Post-pandemic and Energy Crisis Era. In Computational Science and Its Applications – ICCSA 2022 Workshops, Lecture Notes in Computer Science; Gervasi, O.; Murgante, B.; Misra, S, Rocha, A.M.A.C.; Garau, C. Eds.; Springer, Cham. 13380, 345-359 [https://doi.org/10.1007/978-3-031-10542-5_24]
  • Kamargianni, M.; Georgouli, C.; Tronca, L.P.; Chaniotakis, M. Changing transport planning objectives during the Covid-19 lockdowns: Action taken and lessons learned for enhancing sustainable urban mobility planning. Cities 2022, 131, 103873 [https://doi.org/10.1016/j.cities.2022.103873]
  • Maltese, I.; Gatta, V.; Marcucci, E. Active Travel in Sustainable Urban Mobility Plans. An Italian overview. Research in Transportation Business & Management 2021, 40, 100621 [https://doi.org/10.1016/j.rtbm.2021.100621]
  • Zhao, Q.; Xu, H.; Wall, R.S.; Stavropoulos, S. Building a bridge between port and city: Improving the urban competitiveness of port cities. Journal of Transport Geography 2017, 59, 120-133 [https://doi.org/10.1016/j.jtrangeo.2017.01.014]
  • Jugovic, A.; Sirotic, M.; Peronja, I. Sustainable Development of Port Cities from the Perspective of Transition Management. Transactions on Maritime Sciences 2021, 10(2), [https://doi.org/10.7225/toms.v10.n02.w01]
  • Girard, L.F. Toward a Smart Sustainable Development of Port Cities/Areas: The Role of the “Historic Urban Landscape” Approach. Sustainability 2023, 5(10), 4329-4348 [https://doi.org/10.3390/su5104329]
  • Zheng, Y.; Zhao, J.; Shao, G. Port City Sustainability: A Review of Its Research Trends. Sustainability 2020, 12(20), 8355 [https://doi.org/10.3390/su12208355]
  • Seddiek, I.S. Application of renewable energy technologies for eco-friendly sea ports. Ships and Offshore Structures 2020, 15(9), 953-962 [https://doi.org/10.1080/17445302.2019.1696535]
  • Acciaro, M.; Renken, K.; El Khadiri, N. Technological Change and Logistics Development in European Ports. In European Port Cities in Transition: Strategies for Sustainability; Carpenter, A.; Lozano, R. Eds; Springer, 2020, 73-88 [https://doi.org/10.1007/978-3-030-36464-9_5]
  • Tang, J.; McNabola, A.; Misstear, B.; Caulfield, B. An evaluation of the impact of the Dublin Port Tunnel and HGV management strategy on air pollution emissions. Transportation Research Part D: Transport and Environment 2017, 52, 1-14 [https://doi.org/10.1016/j.trd.2017.02.009]
  • D’Amico, G.; Szopik-Depczynsk, K.; Dembinska, I.; Ioppolo, G. Smart and sustainable logistics of Port cities: A framework for comprehending enabling factors, domains and goals. Sustainable Cities and Society 2021, 69, 102801 [https://doi.org/10.1016/j.scs.2021.102801]
  • Port of Bar. The Action Plan – PP7 Port of Bar. Nodes Enhancing Waterway bridging Adriatic-Ionian Network (NEWBRAIN), the project co-funded by ERDF and IPA II fund, 2017 [https://newbrain.adrioninterreg.eu/wp-content/uploads/2021/04/PORT-OF-BAR_Action-plan_NEWBRAIN_signed.pdf]
  • SuMo project (Sustainable Mobility in the Port Cities of the Southern Adriatic Area). Available online: https://sumo.italy-albania-montenegro.eu/ (accessed on December 8th, 2023)
  • Vanoutrive, T. Mobility management in Belgian port areas. In From statics to dynamics: recent advances in network analysis and modelling 2010, NECTAR cluster 1 Meeting on Networks, Antwerpen, Belgium, December 10-11, 2010, [https://hdl.handle.net/10067/852400151162165141]
  • Figliozzi, M.A.; Wei, F.; Jesse, B. Transit Bus Fleet Age and Replacement Type Optimization. OTREC-RR-441. OR: Transportation Research and Education Center (TREC), Portland; 2010. Available online: https://pdxscholar.library.pdx.edu/cgi/viewcontent.cgi?article=1093&context=cengin_fac (accessed on December 8th, 2023).
  • Klimova, A.; Pinho, P. National policies and municipal practices: A comparative study of Czech and Portuguese urban mobility plans. Case Studies on Transport Policies 2020, 8, 1247-1255 [https://doi.org/10.1016/j.cstp.2020.08.005]
  • McDonald, M.; Hall, R.; Hickford, A.; Sammer, G.; Roider, O.; Klementschitz, R. Cluster Report 4: Logistics and Goods Distribution; On behalf of the CIVITAS Initiative, GUARD, European Secretariat, Freiburg, Germany, 2010 [https://civitas.eu/resources/cluster-report-logistics-and-goods-distribution]

Reviewer 3 Report

Comments and Suggestions for Authors

The subject matter of the article is interesting and relevant, although the SUMP methodology has already been extensively researched and described in the literature. Below, I have outlined my suggestions that, in my opinion, could be helpful in improving the article:

  1. The text structure needs improvement. The "Literature Review" subsection should be excluded from the "Materials and Methods" section and should instead be a separate section following the "Introduction."
  2. In the "Literature Review" section, a more in-depth analysis of the literature on sustainable mobility, smart mobility, and sustainable transport systems is needed. It is also valuable to clearly describe the significance of sustainable mobility in the transition to sustainable development in port cities. Specific challenges of sustainable development in port cities could be more detailed.
  3. The literature review presented in subsection 2.1.2 "SUMP in the port cities" should be expanded, as the authors focused on describing implementation projects without conducting a review of scientific research.
  4. In my opinion, the two specific goals of the work, "1) Evaluate the SUMP concept in general and assess the current status and state-of-the-art of SUMPs, including their drivers and barriers; 2) Assess the applicability of the SUMP to the city port by reviewing relevant guides, case studies, and projects," are very ambitious and have not been achieved by the authors - there is a lack of conclusions and recommendations in this regard. This needs to be supplemented or modify the research assumptions in this area.
  5. Regarding "Table 2. The proposed solutions for the port area of Bar," the source of the presented solutions should be indicated. If the solutions presented in Table 2 are proposed by the authors, it should be explained how they were chosen and on what basis they were recommended for achieving the goals outlined in step 3.
  6. The authors point out a lack of interest from stakeholders in participating in the development of a micro-SUMP. It is worthwhile to highlight practices for engaging stakeholders in planning work.
  7. The "Discussion" section is not essentially a scientific discussion, as the authors do not refer to the results of other researchers. The "Discussion" section needs to be redesigned.

I hope these suggestions are helpful in enhancing the quality of the article.

Author Response

To:       Anonymous Reviewer 3

REVISED VERSION

Sustainable Urban Mobility Planning in the Port Areas: A Case Study

Reviewer #3:

Comments to the Author

The subject matter of the article is interesting and relevant, although the SUMP methodology has already been extensively researched and described in the literature. Below, I have outlined my suggestions that, in my opinion, could be helpful in improving the article.

Answer: Thank you very much for kind words regarding the research that we have done. We really appreciate it. Also, thank you for the very valuable comments which further enhance the overall positioning of our research to the wider audience.

Comment 1: The text structure needs improvement. The "Literature Review" subsection should be excluded from the "Materials and Methods" section and should instead be a separate section following the "Introduction."

Answer: Thank you for the comment. We have taken into account your suggestion. "Literature Review" is now a separate new Chapter 2, leading to a renumbering of the subsequent chapters. Additionally, Chapter 3, "Materials and Methods," now consists of two new subchapters: "3.1. Developed Approach" and "3.2. Methodology for creating a territorial SUMP for the port area". This restructuring has enhanced the clarity of the paper, making the content more explicitly described and contextualized in relation to the theoretical background and empirical research.

Comment 2: In the "Literature Review" section, a more in-depth analysis of the literature on sustainable mobility, smart mobility, and sustainable transport systems is needed. It is also valuable to clearly describe the significance of sustainable mobility in the transition to sustainable development in port cities. Specific challenges of sustainable development in port cities could be more detailed.

Answer: Thank you for the comment. Suggestion has been considered, and the "Literature Review" section has been expanded. New references (16, 17, 18, and 19) have been added, linking the terms smart and sustainable mobility with the concept of SUMP. Additionally, new references 26, 27, 28, and 31 have been included to further elucidate the application of SUMP in specific cities. However, in addressing the other reviewer's concern about the theoretical framework being overly detailed, the expansion of the literature analysis was conducted carefully to avoid making the chapter too extensive and potentially compromising the overall paper flow and reader engagement. The specific challenges of sustainable development in port cities are further explained in the text from line 280 to 289 in the Track Changes version). In total, 23 new references have been added (a complete list is provided at the end of the response letter).

Comment 3: The literature review presented in subsection 2.1.2 "SUMP in the port cities" should be expanded, as the authors focused on describing implementation projects without conducting a review of scientific research.

Answer: Thank you very much. We have complied with your request suggestions and the "SUMP in the Port Cities" section has been expanded with new references 38, 39, 40, 41, and 42 (new text from line 265 to line 277 in the Track Changes version). Similar to the previous comment, care has been taken to balance the expansion with the requirements of other reviewers.

Comment 4: In my opinion, the two specific goals of the work, "1) Evaluate the SUMP concept in general and assess the current status and state-of-the-art of SUMPs, including their drivers and barriers; 2) Assess the applicability of the SUMP to the city port by reviewing relevant guides, case studies, and projects," are very ambitious and have not been achieved by the authors - there is a lack of conclusions and recommendations in this regard. This needs to be supplemented or modified by the research assumptions in this area.

Answer: Thank you very much for your valuable comment. In the introductory part of the Chapter 3.2 explaining the proposed methodology (lines 381-382 in the Track Changes version of the paper), the authors stated that the "full implementation processes of the SUMP are beyond the paper’s scope." Specifically, the paper puts forth a methodology with steps to be taken to achieve an optimal micro-SUMP for the port area. The proposed methodology has been validated through a relevant case study, where certain phases of the proposed methodology have been implemented to a greater or lesser extent. Full implementation of these phases would require much more effort and time, surpassing the limitations of this paper. Therefore, the case study is based on a limited implementation of some phases of the proposed methodology, which simultaneously represents a limitation of this research (as later indicated in Chapter 5. Discussion – lines 861-867, as well as Chapter 6. Conclusion – lines 955-960 in the Track Changes version of the paper). To make this clearer, the paper has been supplemented with corresponding statements at the beginning of the Chapter 4. Case Study (lines 416 to 421 in the Track Changes version of the paper), as well as statements in Chapter 4.2 (from line 445 to 461 in the Track Changes version of the paper).

Comment 5: Regarding "Table 2. The proposed solutions for the port area of Bar," the source of the presented solutions should be indicated. If the solutions presented in Table 2 are proposed by the authors, it should be explained how they were chosen and on what basis they were recommended for achieving the goals outlined in step 3.

Answer: Thank you very much. All solutions were jointly proposed by the authors and representatives of the Port of Bar, in consultation with other experts from the SuMo project consortium. Accordingly, appropriate explanations have been added in the Chapter 4.4 (Step 3) from lines 531 to 533 (Track Changes version of the paper), in the Chapter 4.5 (Step 4) lines 567-568 and lines 573-579 (Track Changes version of the paper), as well as in the Chapter 4.7 (Step 6) lines 746-747 (Track Changes version of the paper).

Comment 6: The authors point out a lack of interest from stakeholders in participating in the development of a micro-SUMP. It is worthwhile to highlight practices for engaging stakeholders in planning work.

Answer: We appreciate your suggestion.  We have amended our paper, so in the Chapter 5. Discussion (lines 871-885 in the Track Changes version of the paper), the issue of stakeholder participation and the proposed methods of their involvement are discussed.

Comment 7: The "Discussion" section is not essentially a scientific discussion, as the authors do not refer to the results of other researchers. The "Discussion" section needs to be redesigned.

Answer: Thank you very much for your valuable comment. The entire Discussion chapter has now been redesigned to adhere more closely to scientific conventions.

Comment 8: I hope these suggestions are helpful in enhancing the quality of the article.

Answer: We extend our sincere gratitude to the reviewer for the thoughtful and constructive suggestions provided. Your insights have proven invaluable in refining the quality of our article. We appreciate the time and effort you dedicated to reviewing our work, and your feedback has played a crucial role in enhancing its overall merit.

We hope that all the amendments discussed above meet all your concerns with regards to the content of this work and its presentation.

List of added references

  • European Commission. Communication from the Commission to the European Parliament, the Council, the European Economic and Social Committee and the Committee of the Regions-Together towards competitive and resource-efficient urban mobility. White paper {SWD(2013) 524 final-529 final}, Brussels 2013. [https://eur-lex.europa.eu/resource.html?uri=cellar:82155e82-67ca-11e3-a7e4-01aa75ed71a1.0011.02/DOC_3&format=PDF]
  • Lozano, D.L.A; Marquez, S.E.D.; Puentes, M.E.M. Sustainable and smart mobility evaluation since citizen participation in responsive cities. Transportation Research Procedia 2021, 58, 519-526 [https://doi.org/10.1016/j.trpro.2021.11.069]
  • Benevolo, C.; Dameri, R.P.; D’Auria, B. Smart mobility in smart city. In Empowering Organizations, Lecture Notes in Information Systems and Organisation; Torre, T.; Braccini, A.; Spinelli, R. Eds.; Springer, 2016, 11, 13-28 [https://doi.org/10.1007/978-3-319-23784-8_2]
  • Lyons, G. Getting smart about urban mobility – Aligning the paradigms of smart and sustainable. Transportation Research Part A 2018, 115, 4-14 [https://doi.org/10.1016/j.tra.2016.12.001]
  • Melkonyan, A.; Gruchmann, T.; Lohmar, F.; Bleischwitz, R. Decision support for sustainable urban mobility: A case study of the Rhine-Ruhr area. Sustainable Cities and Society 2022, 80, 103806 [https://doi.org/10.1016/j.scs.2022.103806]
  • Mozos-Blanco, M.A.; Pozo-Menendez, E.; Arce-Ruiz, R.; Baucells-Aleta, N. The way to sustainable mobility. A comparative analysis of sustainable mobility plans in Spain. Transport Policy 2018, 72, 45-54 [https://doi.org/10.1016/j.tranpol.2018.07.001]
  • Charradi, I.; Campisi, T.; Tesoriere, G.; Abdallah, K.B. A Holistic Approach to SUMP Strategies and Actions in the Post-pandemic and Energy Crisis Era. In Computational Science and Its Applications – ICCSA 2022 Workshops, Lecture Notes in Computer Science; Gervasi, O.; Murgante, B.; Misra, S, Rocha, A.M.A.C.; Garau, C. Eds.; Springer, Cham. 13380, 345-359 [https://doi.org/10.1007/978-3-031-10542-5_24]
  • Kamargianni, M.; Georgouli, C.; Tronca, L.P.; Chaniotakis, M. Changing transport planning objectives during the Covid-19 lockdowns: Action taken and lessons learned for enhancing sustainable urban mobility planning. Cities 2022, 131, 103873 [https://doi.org/10.1016/j.cities.2022.103873]
  • Maltese, I.; Gatta, V.; Marcucci, E. Active Travel in Sustainable Urban Mobility Plans. An Italian overview. Research in Transportation Business & Management 2021, 40, 100621 [https://doi.org/10.1016/j.rtbm.2021.100621]
  • Zhao, Q.; Xu, H.; Wall, R.S.; Stavropoulos, S. Building a bridge between port and city: Improving the urban competitiveness of port cities. Journal of Transport Geography 2017, 59, 120-133 [https://doi.org/10.1016/j.jtrangeo.2017.01.014]
  • Jugovic, A.; Sirotic, M.; Peronja, I. Sustainable Development of Port Cities from the Perspective of Transition Management. Transactions on Maritime Sciences 2021, 10(2), [https://doi.org/10.7225/toms.v10.n02.w01]
  • Girard, L.F. Toward a Smart Sustainable Development of Port Cities/Areas: The Role of the “Historic Urban Landscape” Approach. Sustainability 2023, 5(10), 4329-4348 [https://doi.org/10.3390/su5104329]
  • Zheng, Y.; Zhao, J.; Shao, G. Port City Sustainability: A Review of Its Research Trends. Sustainability 2020, 12(20), 8355 [https://doi.org/10.3390/su12208355]
  • Seddiek, I.S. Application of renewable energy technologies for eco-friendly sea ports. Ships and Offshore Structures 2020, 15(9), 953-962 [https://doi.org/10.1080/17445302.2019.1696535]
  • Acciaro, M.; Renken, K.; El Khadiri, N. Technological Change and Logistics Development in European Ports. In European Port Cities in Transition: Strategies for Sustainability; Carpenter, A.; Lozano, R. Eds; Springer, 2020, 73-88 [https://doi.org/10.1007/978-3-030-36464-9_5]
  • Tang, J.; McNabola, A.; Misstear, B.; Caulfield, B. An evaluation of the impact of the Dublin Port Tunnel and HGV management strategy on air pollution emissions. Transportation Research Part D: Transport and Environment 2017, 52, 1-14 [https://doi.org/10.1016/j.trd.2017.02.009]
  • D’Amico, G.; Szopik-Depczynsk, K.; Dembinska, I.; Ioppolo, G. Smart and sustainable logistics of Port cities: A framework for comprehending enabling factors, domains and goals. Sustainable Cities and Society 2021, 69, 102801 [https://doi.org/10.1016/j.scs.2021.102801]
  • Port of Bar. The Action Plan – PP7 Port of Bar. Nodes Enhancing Waterway bridging Adriatic-Ionian Network (NEWBRAIN), the project co-funded by ERDF and IPA II fund, 2017 [https://newbrain.adrioninterreg.eu/wp-content/uploads/2021/04/PORT-OF-BAR_Action-plan_NEWBRAIN_signed.pdf]
  • SuMo project (Sustainable Mobility in the Port Cities of the Southern Adriatic Area). Available online: https://sumo.italy-albania-montenegro.eu/ (accessed on December 8th, 2023)
  • Vanoutrive, T. Mobility management in Belgian port areas. In From statics to dynamics: recent advances in network analysis and modelling 2010, NECTAR cluster 1 Meeting on Networks, Antwerpen, Belgium, December 10-11, 2010, [https://hdl.handle.net/10067/852400151162165141]
  • Figliozzi, M.A.; Wei, F.; Jesse, B. Transit Bus Fleet Age and Replacement Type Optimization. OTREC-RR-441. OR: Transportation Research and Education Center (TREC), Portland; 2010. Available online: https://pdxscholar.library.pdx.edu/cgi/viewcontent.cgi?article=1093&context=cengin_fac (accessed on December 8th, 2023).
  • Klimova, A.; Pinho, P. National policies and municipal practices: A comparative study of Czech and Portuguese urban mobility plans. Case Studies on Transport Policies 2020, 8, 1247-1255 [https://doi.org/10.1016/j.cstp.2020.08.005]
  • McDonald, M.; Hall, R.; Hickford, A.; Sammer, G.; Roider, O.; Klementschitz, R. Cluster Report 4: Logistics and Goods Distribution; On behalf of the CIVITAS Initiative, GUARD, European Secretariat, Freiburg, Germany, 2010 [https://civitas.eu/resources/cluster-report-logistics-and-goods-distribution]

Round 2

Reviewer 2 Report

Comments and Suggestions for Authors

the manuscript still has some grammatical errors and typos. 

It is advisable to have a copyright or permission to use images not created by the authors .

 Once this has been corrected, the paper will be eligible for publication

Comments on the Quality of English Language

Moderate editing of English language required

Author Response

To:       Anonymous Reviewer 2

Reviewer #2:

Comments to the Author

Comment 1: The manuscript still has some grammatical errors and typos.

Answer: The revised manuscript has been proofread again by a native competent English professor. Please point out any remaining grammatical or typo errors.

Comment 2: It is advisable to have a copyright or permission to use images not created by the authors.

Answer: The authors secured permissions from the copyright owner for two figures not authored by them. These permissions are detailed in separate document

Comments on the Quality of English Language: Moderate editing of English language required.

Answer: We appreciate your comment very much. The English language is modified and checked again by native competent English professor.

Reviewer 3 Report

Comments and Suggestions for Authors

Thank you for incorporating the suggested corrections into the text. I wish a high level of interest in the article and numerous citations of this work by other researchers.

Author Response

To:       Anonymous Reviewer 3

Reviewer #3:

Comments to the Author

Comment 1: Thank you for incorporating the suggested corrections into the text. I wish a high level of interest in the article and numerous citations of this work by other researchers.

Answer: Thank you for your positive feedback on the revised version we submitted. We really appreciate it. In addition, we are grateful for your previous valuable comments, as they significantly contributed to strengthening the quality of our research.